# Revisit, Extend, and Enhance Hessian-Free Influence Functions

**Ziao Yang\***                                                    *ziaoyang@brandeis.edu*
*Department of Computer Science*
*Brandeis University*

**Han Yue\***                                                      *hanyue@brandeis.edu*
*Department of Computer Science*
*Brandeis University*

**Jian Chen**                                               *chenj@sem.tsinghua.edu.cn*
*Department of Management Science and Engineering*
*Tsinghua University*

**Hongfu Liu**                                                *hongfuliu@brandeis.edu*
*Department of Computer Science*
*Brandeis University*

**Reviewed on OpenReview:** *https://openreview.net/forum?id=ijL2681Tau*
*\* Equal contribution*

## Abstract

Influence functions serve as crucial tools for assessing sample influence. By employing the first-order Taylor expansion, sample influence can be estimated without the need for expensive model retraining. However, applying influence functions directly to deep models presents challenges, primarily due to the non-convex nature of the loss function and the large size of model parameters. This difficulty not only makes computing the inverse of the Hessian matrix costly but also renders it non-existent in some cases. In this paper, we revisit a Hessian-free method, which substitutes the inverse of the Hessian matrix with an identity matrix, and offer deeper insights into why this straightforward approximation method is effective. Furthermore, we extend its applications beyond measuring model utility to include considerations of fairness and robustness. Finally, we enhance this method through an ensemble strategy. To validate its effectiveness, we conduct experiments on synthetic data and extensive evaluations on noisy label detection, sample selection for large language model fine-tuning, and defense against adversarial attacks.

## 1 Introduction

Data-centric learning is a growing research field that focuses on enhancing machine learning model performance by refining the quality and characteristics of training data (Oala et al., 2023). In contrast to model-centric approaches, which prioritize improving algorithms or optimization techniques, data-centric learning involves adjusting the dataset—through trimming, relabeling, and reweighting—while keeping the learning algorithm fixed. This approach plays a vital role in areas such as model interpretability, selecting training subsets, generating synthetic data, detecting noisy labels, improving active learning, and promoting fairness (Chhabra et al., 2024; Kwon et al., 2023).

Sample influence estimation, as the foundation of data-centric learning, can be generally categorized into two categories (Hammoudeh & Lowd, 2022). (a) Retraining-based methods assess the sample influence by retraining the model with and without a specific sample and checking the performance change, which

include the classical leave-one-out influence approach (Cook & Weisberg, 1982) and Shapley value approaches (Ghorbani & Zou, 2019; Jia et al., 2019; Kwon & Zou, 2022; Jia et al., 2018). (b) Gradient-based methods estimate influence without expensive overheads of retraining, known as influence functions. The seminal work in this category is that of (Koh & Liang, 2017), which utilizes a Taylor-series approximation and LiSSA optimization (Agarwal et al., 2017) to compute sample influences with the inverse of the Hessian matrix and sample gradient. However, the limiting assumption is that the model and loss function are convex. Despite debates on the necessity of convexity (Bae et al., 2022; Grosse et al., 2023; Basu et al., 2020; Epifano et al., 2023), challenges persist when directly applying gradient-based methods to large models. The size of model parameters complicates calculations, particularly in obtaining the inverse of the Hessian matrix. Efforts, including matrix decomposition techniques (Koh & Liang, 2017; Grosse et al., 2023; Kwon et al., 2023), aim to expedite and approximate Hessian matrix inversion, therefore enhance the feasibility of influence functions for deep models in practical applications and complex computations.

**Contributions**. In this paper, we focus on the influence function category and revisit a specific naive yet aggressive approximation method, TracIn (Pruthi et al., 2020) or an early version (Charpiat et al., 2019). This method substitutes the inverse of the Hessian matrix with an identity matrix, representing it as a Hessian-free influence function—the inner product of the gradient of the validation set and a sample to be assessed, which we refer to as Inner Product (IP). Rather than proposing a novel algorithm, our paper delivers an important message to the data-centric community: existing Hessian-free, simple methods, such as IP, might outperform more complex approaches that rely on dedicated Hessian inverse approximations. And we further explain the underlying reason. This finding emphasizes the importance of simplicity, efficiency, and scalability in influence estimation methods. Furthermore, it highlights the potential of these straightforward techniques to serve as a robust foundation for extending a wide range of data-centric diverse tasks. We summarize our major contributions as follows:

- We revisit the Hessian-free influence function, define it into IP formulation, and delve into the rationale behind this simple approximation, offering insights into why it performs well in practice.

- Expanding our IP framework, we extend its applicability beyond measuring sample influence on model utility to fairness and robustness.

- To enhance the generalization, we propose IP Ensemble, a novel approach leveraging dropout mechanisms to simulate diverse models. IP Ensemble amalgamates IP scores from these varied models, thus increasing the method's generalization capabilities.

- We validate the effectiveness of IP through synthetic data experiments and conduct extensive real-world evaluations using IP Ensemble. These experiments span various applications, including noisy label correction for vision data, data curation aimed at fine-tuning fairer NLP models, and defense strategies against adaptive evasion adversaries.

## 2 Related Work

In this section, we introduce the literature on influence functions, with a focus on the acceleration of the calculation of the inverse of Hessian matrix, followed by various applications and miscellaneous.

**Efficient Influence Estimation.** Influence functions serve as crucial tools for estimating the individual valuation of data without requiring model retraining. However, the computation of the inverse of the Hessian matrix poses challenges for large-scale data and models. To address this issue, various approaches have been proposed to simplify or estimate the inverse of the Hessian matrix effectively. A seminal work is that of (Koh & Liang, 2017), which employs a Taylor-series approximation and LiSSA optimization (Agarwal et al., 2017) to compute sample influences. Arnoldi (Schioppa et al., 2022) employs the random projection and simplified Hessian matrix for acceleration. EKFAC (Grosse et al., 2023) enhances Kronecker-Factored eigendecomposition for a precise Hessian approximation. More recently, DataInf (Kwon et al., 2023) efficiently computes influence even for large models by replacing the inverse Hessian computation with a readily computable closed-form expression, although their framework may suffer from significant theoretical errors. TracInc (Pruthi et al., 2020), a straightforward yet aggressive approximation, substitutes the inverse

of the Hessian matrix with an identity matrix, essentially considering gradients directly as a measure of influence. Beyond the conventional influence function that gauges sample influence on the validation set, self-influence (Bejan et al., 2023; Thakkar et al., 2023) computes influence using the training set alone. Moving beyond using a single model checkpoint, GEX (Kim et al., 2024) leverages a geometric ensemble of multiple checkpoints to approximate influence functions, alleviating the bilinear constraint and non-linear losses. Moreover, TDA (Bae et al., 2024) also introduces a checkpoint-based segmentation approach, combining implicit differentiation and unrolling by using EKFAC (Grosse et al., 2023). One concurrent work Deng et al. (2024) employs ensemble strategy to improve the efficiency of TRAK Park et al. (2023), a random project-based method to tackle high-dimensionaly to the Hessian matrix.

**Various Applications of Influence Functions**. With the above efficient approximation, influence functions have diverse applications. One major application is identifying detrimental samples (Hammoudeh & Lowd, 2024). The learning performance can be further improved by removing (Chhabra et al., 2024), relabeling (Kong et al., 2021), or reweighting (Thakkar et al., 2023) the identified detrimental samples, which has significant implications in various fields such as noisy label detection (Wang et al., 2020), subset selection (Ting & Brochu, 2018), and the identification of the most influential samples (Sharchilev et al., 2018; Xia et al., 2024). Other applications encompass few-shot learning (Park et al., 2021), where influence functions help improve model performance with minimal data, and recommendation systems (Li et al., 2023; Zhang et al., 2023), enhancing the accuracy and personalization of recommendations. Influence functions are also valuable in selecting data for active learning (Liu et al., 2021), fairness machine learning (Li & Liu, 2022; Wang et al., 2022; 2024), adversarial attack (Cohen et al., 2020), graph machine learning (Chen et al., 2023; Wu et al., 2023), machine unlearning (Xu et al., 2024; Tarun et al., 2023), out-of-distribution generalization (Ye et al., 2021), data privacy (Carey et al., 2023), domain adaptation (Zhang et al., 2022), to name a few. Overall, influence functions are a powerful tool with a wide range of applications across different domains, contributing to advancements in both theoretical and practical aspects of machine learning.

**Miscellaneous**. Several studies have examined the fragility of influence functions in explaining deep learning model predictions. Koh et al. (2019) expands influence functions from estimating the effects of removing one point to large groups of training samples. Chen et al. (2020) extend traditional influence functions to monitor the impact of pre-training data on fine-tuned model predictions, facilitating the identification of crucial examples. Basu et al. (2020) demonstrates that the effectiveness of influence functions in neural networks varies with network architecture, depth, width, parameterization, and regularization, underscoring their fragility in deep learning due to non-convex loss functions. Bae et al. (2022) discovers that while influence estimates may not perfectly align with leave-one-out retraining, they approximate the proximal Bregman response function, offering valuable insights for identifying influential or mislabeled examples. Epifano et al. (2023) suggests that the instability of current validation procedures, rather than non-convexity or lack of regularization, may be responsible for their unreliability. Lyu et al. (2023) enhance influence estimation in large-scale models by concentrating on target parameters and addressing computational instability with a robust inverse-Hessian-vector product approximation.

## 3 Methods

In this section, we introduce the preliminaries of the influence function, with a focus on the Hessian-free approximation, then elaborate on our extension and upgrade.

**Revisit and Simplify**. Given a training set $T=\{z_i=(x_i, y_i)\}_{i=1}^n$ and a classifier with empirical risk minimization by a convex loss function $\ell$, the optimal parameters of the classifier can be obtained by $\hat{\theta} = \arg\min_{\theta \in \Theta} \frac{1}{n} \sum_{i=1}^n \ell(z_i; \theta)$. To measure the influence of an individual data sample, we can train the model with and without the specific sample and see the performance change. However, the retrain-based approach is expensive for large-scale data and models. To avoid model retraining, influence functions estimate the effect of changing an infinitesimal weight of samples on a validation set $V=\{z_j=(x_j, y_j)\}_{j=1}^{n'}$, based on an impact function $f$ evaluating the quantity of interest. Considering the sample impact on model utility, *i.e.,* the loss on the validation set, by removing one sample from the training set, this sample influence can be estimated as follows (Koh & Liang, 2017):

$$\mathcal{I}^{\text{util}}(-z_i) = \sum\nolimits_{z_j \in V} \nabla_{\hat{\theta}} \ell(z_j; \hat{\theta})^\top \mathbf{H}_{\hat{\theta}}^{-1} \nabla_{\hat{\theta}} \ell(z_i; \hat{\theta}), \tag{1}$$

where $\mathbf{H}_{\hat{\theta}} = \sum_{i=1}^{n} \nabla_{\hat{\theta}}^2 \ell(z_i; \hat{\theta})$ is the Hessian matrix of the convex $\ell$ loss function.

Influence functions encounter a challenge for deep models, primarily due to the non-convex nature of the loss function and the considerable size of model parameters. This obstacle not only renders the calculation of the inverse of the Hessian matrix costly but also leads to its non-existence. Various attempts, including matrix decomposition methods (Koh & Liang, 2017; Grosse et al., 2023; Kwon et al., 2023), have been undertaken to expedite and approximate the inversion of the Hessian matrix, aiming to render influence functions viable for deep models. In this paper, we revisit a particular naive yet aggressive approximation method TracIn (Pruthi et al., 2020) or an early version Charpiat et al. (2019) by substituting the inverse of the Hessian matrix with an identity matrix, outlined as follows:

$$\mathcal{I}_{\text{IP}}^{\text{util}}(-z_i) = \sum_{z_j \in V} \nabla_{\hat{\theta}} \ell(z_j; \hat{\theta})^{\top} \cdot \nabla_{\hat{\theta}} \ell(z_i; \hat{\theta}). \tag{2}$$

The above calculation can be regarded as the inner product of two gradients; therefore, we call this method *Inner Product (IP)*. Note that TracIn (Pruthi et al., 2020) incorporates multiple checkpoints to record model parameters throughout the optimization process, whereas Charpiat et al. (2019) or IP only takes into account the final or converged model. While it seems appealing to consider the dynamic sample influence change according to the optimized model parameters, simply summing these simple influence scores along different checkpoints fails to account for the evolving nature of influence, potentially neutralizing conflicting values and reducing overall effectiveness, which can be verified in our experimental results (See Table 1).

In the following, we link the IP to influence functions and provide our insights of IP on why such a naive approximation works well in practice from the angle of both the traditional influence function and IP. Traditional influence functions and their approximations require the inverse of the Hessian matrix. However, the non-linear nature of data and models not only makes this computation challenging but may also lead to the non-existence of a Hessian inverse. To address this, a modified Hessian matrix $(\mathbf{H}_{\hat{\theta}} + \lambda \mathbf{I})^{-1}$ is often employed, with approximations Koh & Liang (2017); Grosse et al. (2023); Kwon et al. (2023) from matrix theory used to calculate the influence score. However, choosing the regularization parameter $\lambda$ presents a tradeoff: a small $\lambda$ may not guarantee the existence of the inverse, while a large $\lambda$ makes the inverse approximate the identity matrix. From the perspective of IP, it measures the similarity between two samples—the overall validation set and a specific training sample. A high IP score indicates that the target training sample is similar to the validation set, suggesting that this sample contributes positively to improving the model's performance for both convex and non-convex models. However, in modern deep networks the Hessian is typically large, ill-conditioned, and often indefinite, so $H_{\hat{\theta}}^{-1}$ may not exist in a strict sense and must be heavily regularized. As a result, Hessian-inverse approximations can be numerically unstable, and their approximation error is difficult to quantify, especially in non-convex settings. In this work we therefore adopt the Hessian-free inner product in Eq. (2): rather than aiming to faithfully approximate $H_{\hat{\theta}}^{-1}$, we focus on a simple and scalable gradient-alignment signal between training samples and a validation set. While IP may not always recover the exact influence value, it provides a reliable indication of the polarity of influence (beneficial vs. detrimental), which is often the main quantity needed in data-centric applications.

**Extension**. Beyond measuring sample influence on model utility, we extend IP to assess the sample influence on fairness and robustness by modifying the impact function $f$.

Specifically, we can instantiate the impact function $f$ by group fairness (Dwork et al., 2012), such as demographic parity (DP) to measure influence on fairness (Li & Liu, 2022). Consider a binary sensitive attribute defined as $g \in \{0, 1\}$ and let $\hat{y}$ denote the predicted class probabilities. The fairness metric DP is defined as: $f^{\text{DP-fair}}(\hat{\theta}, V) = \left| \mathbb{E}_V[\hat{y}|g=1] - \mathbb{E}_V[\hat{y}|g=0] \right|$. Within the above IP framework, we can calculate the training sample influence on fairness as follows:

$$\mathcal{I}_{\text{IP}}^{\text{DP-fair}}(-z_i) = \nabla_{\hat{\theta}} f^{\text{DP-fair}}(V; \hat{\theta})^{\top} \cdot \nabla_{\hat{\theta}} \ell(z_i; \hat{\theta}). \tag{3}$$

Similarly, we can also measure the sample influence on adversarial robustness within the IP framework. To achieve this, we follow Chhabra et al. (2024) and consider a white-box adversary (Megyeri et al., 2019) specific to linear models, which can be easily extended to other models and settings. To craft an adversarial sample, we take each sample $z_j = (x_j, y_j)$ in the validation set $V$ and only perturb $x_j' = x_j - \gamma \frac{\hat{\theta}^{\top} x_j + b}{\hat{\theta}^{\top} \hat{\theta}} \hat{\theta}$ and

keep $y_j$ unchanged, where $\hat{\theta} \in \mathbb{R}^d$ are the linear model coefficients, $b \in \mathbb{R}$ is the intercept, and $\gamma > 1$ controls the amount of perturbation added. In this manner, we can obtain an adversarial validation set $V'$ which consists of $z'_j = (x'_j, y_j)$ for each sample $z_j$ of $V$. Now, we can compute adversarial robustness influence for each training sample as follows:

$$\mathcal{I}_{\text{IP}}^{\text{robust}}(-z_i) = \sum_{z'_j \in V'} \nabla_{\hat{\theta}} \ell(z'_j; \hat{\theta})^\top \cdot \nabla_{\hat{\theta}} \ell(z_i; \hat{\theta}). \tag{4}$$

**Enhancement**. The simplicity of IP offers opportunities to enhance the generalization of influence functions. In convex cases, the model parameter $\hat{\theta}$ is both optimal and unique. However, in non-convex scenarios, the presence of local minima introduces instability and non-uniqueness into the solution. Typically, ensemble strategies are employed to bolster model generalization (Dietterich, 2000; Lakshminarayanan et al., 2017). Yet, while employing different models can enhance performance, it also escalates the costs associated with model training and complicates the calculation of influence functions. This arises from the variability in model parameters, necessitating multiple computations of the inverse of each individual Hessian matrix. The introduction of Hessian-free IP circumvents this issue, eliminating the need for costly calculations of Hessian matrices and their inverses. Drawing inspiration from dropout mechanisms, diverse models can be swiftly generated without necessitating model retraining. By computing sample gradients from various models, we propose IP Ensemble that amalgamates IP scores from distinct models. Experiments detailed in Section 5 illustrate superior performance of IP Ensemble over other influence function-based methods.

**When can IP fail?** Despite the above insights and enhancements, IP is not universally reliable. IP can be viewed as a Hessian-free approximation of classical influence functions, where the inverse Hessian in Eq. (1) is replaced by the identity matrix, yielding Eq. (2). Under Eq. (2), IP essentially measures the alignment between a training sample's gradient and the validation loss gradient (i.e., the aggregated gradient over the validation set). It is worthy noting that influence functions are not accurate tools for data valuation in the deep models. Fortunately, it holds for deep models that adding more validation-like training samples tends to reduce the validation loss, which typically holds when the model is sufficiently trained and the validation gradient provides a stable direction of improvement. However, when the model is ill-trained or under-trained (e.g., far from convergence), gradients can be noisy or unstable, and the validation gradient may not yet be informative about the final optimization geometry. In such cases, the gradient-alignment signal can become unreliable, and IP may fail to rank samples correctly.

## 4 Correctness Verification on Synthetic Data

We evaluate the effectiveness of our Inner Product (IP) score as a surrogate for influence functions on two synthetic datasets representing convex and non-convex optimization settings.

For the convex case, we generate a linearly separable dataset using scikit-learn `make_blobs` function, consisting of 150 training samples and 100 test samples. We manually flip the labels of 10 training samples to introduce noise. A logistic regression model is trained on this dataset to assess IP under convex optimization. For the non-convex case, we use the *half-moons* dataset generated by `make_moons`, with 200 training samples (including 20 label-flipped noisy samples, 10 per class) and 100 test samples. We train a three-layer Multi-Layer Perceptron (MLP) with fully connected layers and ReLU activations. The network maps the input to 32 hidden units, maintains this dimensionality in the second layer, and outputs a single sigmoid-activated neuron for binary prediction.

Figure 1**A**–**B** visualize the linear dataset, while **D**–**E** present the half-moons dataset. Figures 1 **C** and **F** report the influence scores and IP scores for each training sample, computed using Eqs. (1) and (2). In the convex setting, the IP score closely matches the influence score, showing near-perfect correlation and order consistency. Detrimental samples receive negative scores under both measures, whereas beneficial or neutral samples yield positive or near-zero values. In contrast, in the non-convex setting, influence scores become less reliable: detrimental samples are intermingled with normal samples, likely due to inaccuracies in Hessian inverse approximation. The IP score, however, continues to clearly separate detrimental samples from inliers. As discussed previously, a low IP score reflects strong divergence between a training sample and the validation objective, indicating negative contribution. Notably, despite its simplicity, IP performs

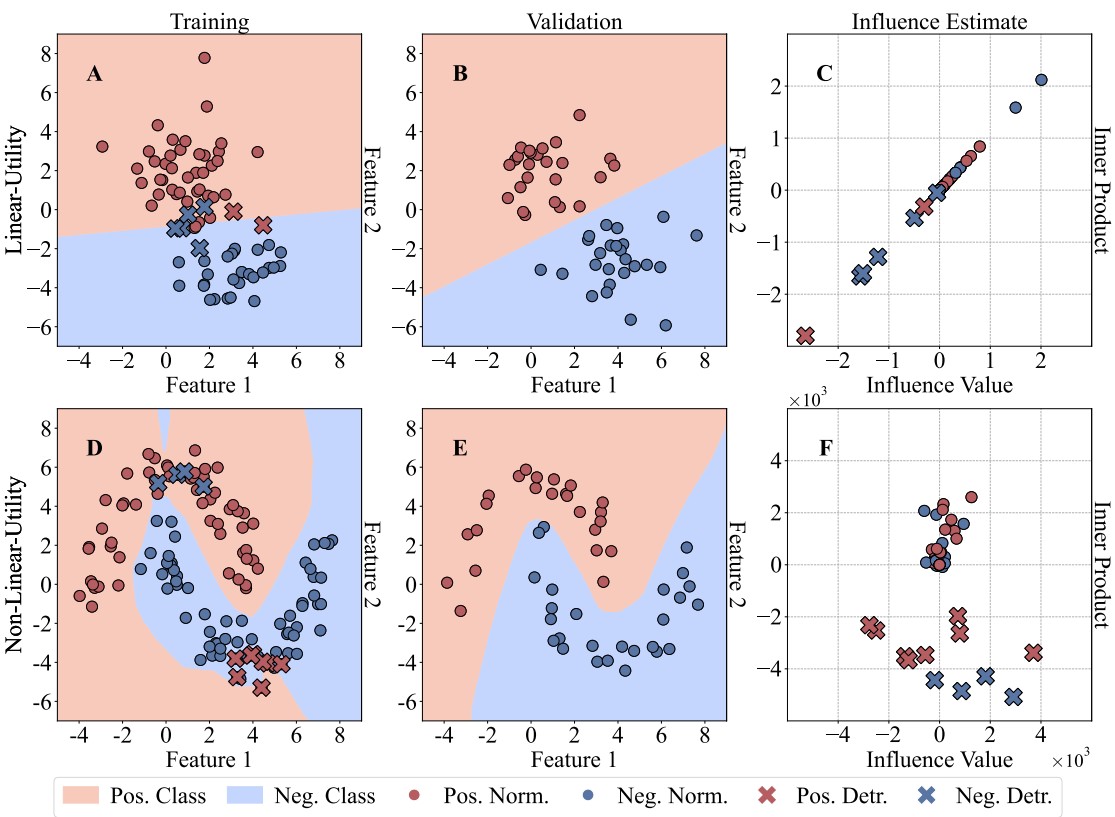

Figure 1: Illustration of IP on two synthetic datasets and convex/non-convex models. **A-C** illustrate a 2D linearly separable synthetic dataset with a subset of detrimental samples bearing incorrect labels, and **D-F** demonstrate the similar analysis on a non-linear synthetic half-moon dataset. Specifically, **A** and **D** depict training sets with two classes, where detrimental samples are marked with × and regular samples with ○. **B** and **E** show test sets. **C** and **F** present influence scores and IP scores by Eqs. (1) and (2), respectively.

comparably to more complex methods that rely on Hessian-inverse approximations, even in the non-convex case. Additional empirical evaluations on model fairness and robustness are provided in Appendix A.1.

Next, we will present a unified empirical program along three sections: (1) noisy-label data curation for vision, where we rank samples by influence, and retrain to measure accuracy gains; (2) fairness-aware curation for NLP, where we compute influence on both utility and a demographic-perturbation fairness score, and visualize the accuracy-fairness frontier; (3) robustness to adaptive test-time evasion, where we compare pre/post-attack accuracy and evaluate three defenses.

## 5    Noisy Label Correction for Vision Datasets

In this section, we demonstrate the effectiveness of our IP Ensemble in identifying detrimental samples on noisy vision datasets. Specifically, we choose three benchmark datasets *CIFAR-10N* (Wei et al., 2022), *CIFAR-100N* (Wei et al., 2022), and *Animal-10N* (Shu et al., 2023) in the noisy label learning area. Both the *CIFAR-10N* and *CIFAR-100N* datasets (Oliver et al., 2018) consist of the same input images as their clean counterparts, *CIFAR-10* (10 classes) and *CIFAR-100* (100 classes) (Krizhevsky et al., 2009), respectively. Each input is a 32x32 RGB image with dimensions (3,32,32). However, for *CIFAR-10N* and *CIFAR-100N*, the labels contain real-world human annotation errors collected using three annotators on Amazon Mechanical Turk. Since these datasets are based on human-annotated noise, they provide a more realistic modeling of noisy real-world scenarios compared to synthetic alternatives. The training set for both datasets contains 50,000 image-label pairs, while the test set contains 10,000 clean image-label pairs. Specifically, *CIFAR-*

*10N* encompasses three distinct noise settings: aggregate ("-*a*"), random ("-*r*"), and worst ("-*w*"). The "*aggregate*" setting has the lowest noise rate (9.03%), where labels are determined via majority voting among three annotators, with ties resolved randomly. The "*random*" setting has intermediate noise (17.23%), adopting labels from the first annotator. The "*worst*" setting has the highest noise rate (40.21%), deliberately selecting the most inconsistent annotation per sample.

For competitive methods, we choose the following influence function-based methods. TracIn (Pruthi et al., 2020) replaces the Hessian matrix with the identity matrix and considers checkpoints during the training process; LiSSA (Koh & Liang, 2017) and EKFAC (Grosse et al., 2023) employ implicit Hessian-vector products and Kronecker-Factored curvature to efficiently approximate the inverse of the Hessian matrix; DataInf (Kwon et al., 2023) swaps the order of matrix multiplication for obtaining a closed-form expression; Self-TracIn (Thakkar et al., 2023) and Self-LiSSA (Bejan et al., 2023) are the self-expression versions of TracIn and LiSSA, where the gradients of the validation set are replaced with $\nabla_{\hat{\theta}}\ell(z_i; \hat{\theta})$, and only the last checkpoint, i.e., the converged model parameters, is used in Self-TracIn. GEX (Kim et al., 2024) utilizes ensemble methods based on checkpoints from extra stochastic gradient descent on the converged model. TDA (Bae et al., 2024) focuses on checkpoints during the training process, ensembling the influence via EKFAC (Grosse et al., 2023). Our IP Ensemble method constitutes an ensemble version of IP with $\mathcal{U}(0, 0.01)$ dropout applied on model parameters and an ensemble size of 5.

TracIn, GEX, IP, and IP Ensemble are all Hessian-free ensemble methods except for IP. To clarify their differences, we present their calculations explicitly:

$$\mathcal{I}_{\text{IP}}^{\text{util}}(-z_i, \hat{\theta}) = \sum\nolimits_{z_j \in V} \nabla_{\hat{\theta}}\ell(z_j; \hat{\theta})^{\top} \nabla_{\hat{\theta}}\ell(z_i; \hat{\theta}),$$

$$\mathcal{I}_{\text{TracIn}}^{\text{util}}(-z_i, \Theta_{\text{TracIn}}) = \frac{1}{T} \sum\nolimits_{\hat{\theta}_t \in \Theta_{\text{TracIn}}} \mathcal{I}_{\text{IP}}^{\text{util}}(-z_i, \hat{\theta}_t),$$

$$\mathcal{I}_{\text{GEX}}^{\text{util}}(-z_i, \Theta_{\text{GEX}}) = \frac{1}{T} \sum\nolimits_{\hat{\theta}_t \in \Theta_{\text{GEX}}} \mathcal{I}_{\text{IP}}^{\text{util}}(-z_i, \hat{\theta}_t),$$

$$\mathcal{I}_{\text{IP Ensemble}}^{\text{util}}(-z_i, \Theta_{\text{IP Ensemble}}) = \frac{1}{T} \sum\nolimits_{\hat{\theta}_t \in \Theta_{\text{IP Ensemble}}} \mathcal{I}_{\text{IP}}^{\text{util}}(-z_i, \hat{\theta}_t),$$

$$(5)$$

where $\hat{\theta}$ in IP represents the converged model parameters; $\Theta_{\text{TracIn}}$ denotes the checkpoints saved during training; $\Theta_{\text{GEX}}$ is derived by additional training on the converged model; IP Ensemble obtains $\Theta_{\text{IP Ensemble}}$ through dropout mechanisms on the converged model; $T$ is the ensemble size. The advantages of IP Ensemble are that it does not require checkpoint saving as in TracIn, nor does it necessitate extra training iterations as in GEX. We argue that early checkpoints may not effectively reflect sample influence, and additional training on converged models has minimal impact. We provide our implementation in an anonymous open-source repository: https://anonymous.4open.science/r/IP_ensemble-BA0F/README.md. The experiments were conducted on a Linux (Ubuntu 20.04.6 LTS) server using NVIDIA GeForce RTX 4090 GPUs with 24GB VRAM running CUDA version 12.3 and driver version 545.23.08.

To reduce randomness, we train a ResNet-34 network (He et al., 2016), which is a 34-layer convolutional neural network pretrained on the ImageNet-1K dataset at resolution $224 \times 224$. The pretrained model block is fine-tuned on the *CIFAR-10N/CIFAR-100N* training set using default parameters: minibatch size (128), optimizer (SGD), initial learning rate (0.1), momentum (0.9), weight decay (0.0005), and number of epochs (100). Subsequently, based on the same model, we employ the aforementioned influence function-based methods to identify 5% of detrimental samples. Following this, we conduct five retraining iterations of the ResNet-34 network, each time removing the identified detrimental samples from the training set. For each method, we repeat the following procedure five times (each time with an independent run): we first identify the bottom 5% most detrimental samples, then retrain the model on the remaining 95% of the training data. Tables 1 and 5 report the results averaged over these five runs. Table 1 reports the average accuracy and standard deviation of the above influence function-based methods across these five retrainings. In general, these influence function-based methods are effective in identifying detrimental samples at different levels. Upon retraining the ResNet-34 model without these identified samples, nearly every result obtained by these methods outperforms the vanilla ResNet-34 trained on the entire dataset, except for EKFAC on *CIFAR-10N-w*. Moreover, TracIn and TDA achieve only moderate performance, largely due to their reliance on

Table 1: Accuracy results of influence function-based methods on the *CIFAR10N*, *CIFAR-100N* and *Animal-10N* datasets with 5% identified detrimental samples removed

| Methods / Datasets | CIFAR-10N-a | CIFAR-10N-r | CIFAR-10N-w | CIFAR-100N | Animal-10N | Avg. |
|---|---|---|---|---|---|---|
| Cross Entropy | 91.62 | 90.25 | 85.66 | 56.41 | 80.54 | 80.90 |
| LiSSA (Koh & Liang, 2017) | 92.13 ± 0.29 | 90.98 ± 0.16 | 85.97 ± 0.47 | 59.24 ± 0.39 | 81.93 ± 0.14 | 82.05 |
| TracIn (Pruthi et al., 2020) | 90.48 ± 0.12 | 88.09 ± 0.24 | 85.18 ± 1.05 | 56.47 ± 1.87 | 80.12 ± 0.57 | 80.07 |
| EKFAC (Grosse et al., 2023) | 91.76 ± 0.23 | 90.47 ± 0.10 | 83.25 ± 0.38 | 59.91 ± 0.90 | 80.89 ± 0.54 | 81.26 |
| DataInf (Kwon et al., 2023) | 91.88 ± 0.39 | 90.79 ± 0.21 | 86.22 ± 0.13 | 58.40 ± 0.22 | 81.60 ± 0.23 | 81.78 |
| Self-TracIn (Thakkar et al., 2023) | 92.03 ± 0.09 | 90.43 ± 0.24 | 86.00 ± 0.18 | 61.99 ± 0.29 | 81.82 ± 0.34 | 82.45 |
| Self-LiSSA (Bejan et al., 2023) | 91.91 ± 0.17 | 90.66 ± 0.35 | 85.73 ± 0.41 | 61.56 ± 0.56 | 81.23 ± 0.24 | 82.22 |
| TDA (Bae et al., 2024) | 91.95 ± 0.19 | 89.87 ± 0.32 | 84.02 ± 0.41 | 58.91 ± 0.48 | 80.57 ± 0.25 | 81.06 |
| GEX (Kim et al., 2024) | 91.81 ± 0.27 | 90.68 ± 0.39 | 85.64 ± 0.20 | 58.47 ± 0.48 | 80.78 ± 0.58 | 81.49 |
| IP (Ours) | **92.42** ± 0.17 | 90.82 ± 0.08 | 86.31 ± 0.35 | 60.59 ± 0.20 | 81.19 ± 0.22 | 82.27 |
| IP Ensemble (Ours) | 92.26 ± 0.19 | **91.28** ± 0.29 | **86.50** ± 0.35 | **62.25** ± 0.54 | **82.35** ± 0.55 | **82.93** |

multiple checkpoints throughout the training process. This highlights why we believe that simply summing up sample influence across different checkpoints fails to effectively capture the dynamic and evolving value of a sample during training.

Notably, our IP Ensemble consistently outperforms other baseline methods across various noise conditions and datasets. Particularly noteworthy is the performance of our IP Ensemble in the most challenging scenario, *CIFAR-100N*, achieving the highest recorded accuracy of 62.25% on the test set, surpassing the vanilla cross-entropy accuracy of 56.41%. Compared to IP, IP Ensemble delivers further performance gains, highlighting the benefits of the ensemble strategy for enhancing model generalization. Moreover, the average accuracy of the IP Ensemble on the test set reaches its peak at 82.93%, surpassing both the vanilla cross-entropy performance of 80.90% and the second-best accuracy of Self-TracIn at 82.45%. More experimental results and analysis on different percentages of removed samples, parameter analysis on different dropout rates, ensemble sizes, and base model architectures can be found in Appendix A.2- A.4.

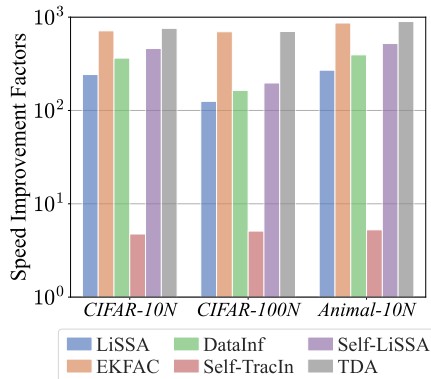

Figure 2: Speed improvement factors of IP over other methods.

In addition, we also present the running times of these influence function-based methods. Despite some baselines having linear time complexity, there is significant divergence in their real execution times. Given that our IP exhibits exceptional speed and similar execution times, we consider them as the baseline and compute the speed improvement factors over other baseline methods, as depicted in Figure 2. For ensemble methods including TracIN, TDA, GEX, and our IP Ensemble, parallel computation can be applied to accelerate the running time if enough resources are allowed; if calculated serially, the time grows linearly with the ensemble size. When we visualize their running time in the logarithm scale, the extended time due to the ensemble size is not significant; for example, EKFAC and its ensemble version TDA. Thus, we do not report the ensemble-based methods in Figure 2, except for the above example of TDA. With the exception of Self-TracIn, our IP runs over 100 times faster than LiSSA, EKFAC, DataInf, and Self-LiSSA. Notably, on *Animal-10N*, IP is over 800 times faster than EKFAC. The time complexities and execution time of these methods can be found in Appendix A.5.

# 6 Data Curation Towards Fine-Tuning of Fairer NLP Models

In this section, we further demonstrate the efficacy of our IP method in gauging the impact of individual samples on fairness within the realm of curating suitable data samples for fine-tuning language models. Beyond mere utility, fairness has emerged as an indispensable attribute for machine learning models to mitigate inadvertent discrimination.

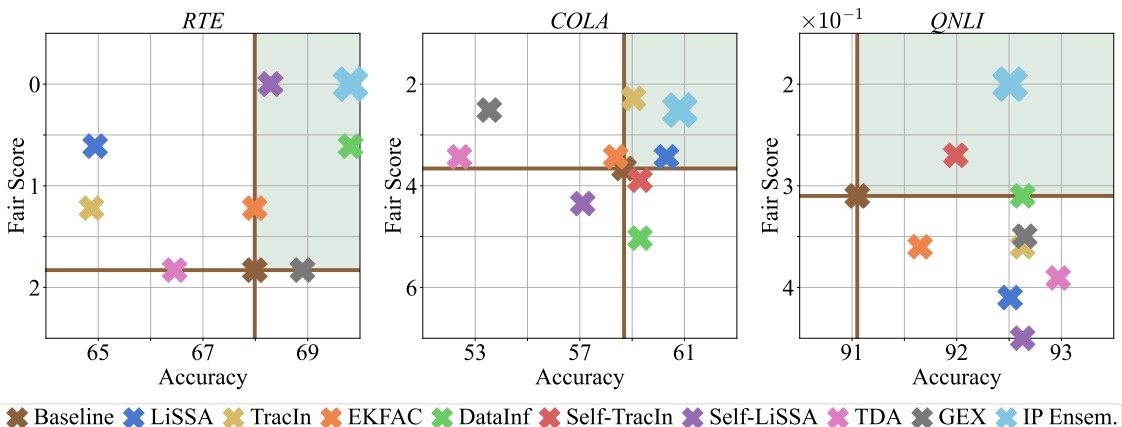

Figure 3: Accuracy and fair score of different influence function-based methods on fine-tuning *RTE*, *COLA*, and *QNLI* datasets. The X-axis denotes accuracy, and the Y-axis for fair score is inverted. The brown crossing denotes the performance of using all the samples for fine-tuning the RoBERTa model as the baseline model. Building upon this, we plot a horizontal and a vertical line in each figure and divide the space by fairness and utility results into four regions. The green area in the top right corner signifies a model that is both fairer and more accurate compared to the baseline model.

In this experiment, we employ three datasets—*RTE*, *CoLA*, and *QNLI*—from the GLUE repository (Wang et al., 2018) to fine-tune the widely-used language model RoBERTa (Liu et al., 2019). Our focus is on group fairness, necessitating that machine learning models treat samples within various predefined subgroups comparably. These datasets represent diverse natural language understanding tasks, thus allowing us to comprehensively evaluate the fairness and utility of the fine-tuned RoBERTa model. The *Recognizing Textual Entailment* (*RTE*) dataset consists of sentence pairs labeled as entailment or non-entailment, derived from a series of textual entailment challenges, containing 2,490 training examples and 277 validation examples. The *Corpus of Linguistic Acceptability* (*CoLA*) dataset includes sentences labeled as grammatically acceptable or unacceptable, derived from linguistic publications, and consists of 8,551 training examples and 1,043 validation examples. The *Question Natural Language Inference* (*QNLI*) dataset is a large-scale corpus for question answering, comprising question-sentence pairs from the Stanford Question Answering dataset, with 104,743 training examples and 5,463 validation examples. As the test sets for these datasets lack labels, we split each validation set into two equal parts, utilizing one half as the validation set for computing influence, and the other half as the test set. Perturbed versions of both the validation and test sets are generated using a seq2seq model as detailed in Qian et al. (2022). To assess fairness, we adopt the methodology outlined in Qian et al. (2022), which involves perturbing the demographic information within each sample and scrutinizing whether the model yields identical predictions for the original sample $x$ and its corresponding perturbed counterpart $\tilde{x}$. The fairness evaluation metric ("fair score") is defined as $|\mathcal{C}(x) - \mathcal{C}(\tilde{x})|$ over all test samples, normalized by the test set size, where $\mathcal{C}(\cdot)$ is the model predictor. Note that the fair score is a negative metric; hence, smaller values are preferable. We fine-tune the RoBERTa-base model from Huggingface[1] with the following experimental settings: a learning rate of $1 \times 10^{-5}$, batch sizes of 64, 16, and 32 for *RTE*, *CoLA*, and *QNLI*, respectively, and a total of 10 epochs. The loss function used is negative log-likelihood, appropriate for these binary classification tasks.

Utility and fairness serve as distinct perspectives for evaluating model performance. Thus, in our fine-tuning experiments, we consider both dimensions. Employing the same influence function-based methods and our IP Ensemble as in the previous section, we conduct comparative analyses. For each method, we compute influence scores for both utility and fairness on the fine-tuning set, aggregate them into a single joint ranking, and remove the bottom 5% samples under this joint criterion before fine-tuning the RoBERTa model.

---

[1]https://huggingface.co/docs/transformers/model_doc/roberta

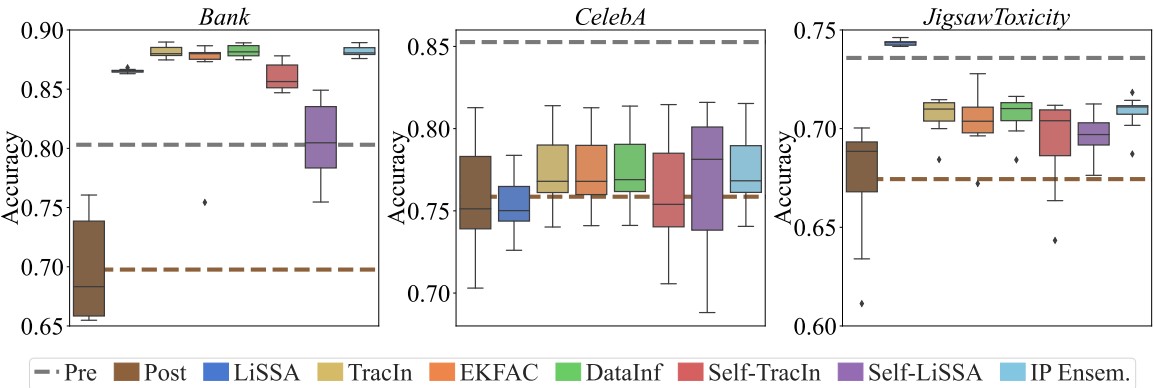

Figure 4: Performance across influence function-based methods over 10 distinct attacks on *Bank*, *CelebA*, and *JigsawToxicity*. The dashed gray line presents the pre-attack performance, while the brown line denotes the average accuracy of post-attack.

The results of this experiment are presented in Figure 3, where the Y-axis for fairness is inverted. The brown crossing denotes the performance of using all the samples for model fine-tuning as the base model. Building upon this, we plot a horizontal and a vertical line in each figure and divide the space by fairness and utility results into four regions. The green area in the top right corner signifies a model that is both fairer and more accurate compared to the baseline model. Results within this green region can be considered Pareto improvements, enhancing both utility and fairness simultaneously. It is evident most results are located in the green area, indicating the existence of detrimental samples, and not all the samples are helpful to the model performance. This also implies that influence function methods are effective to identifying the detrimental samples, even for non-convex deep models. However, some competitive methods yield a trade-off or even Pareto deterioration. For instance, LiSSA demonstrates a better fair score but worse accuracy compared to the base model on *RTE*; conversely, it exhibits better accuracy but worse fairness on *QLIN*. EKFAC shows similar performance on *COLA* and *QLIN*. Self-LiSSA demonstrates Pareto deterioration on *COLA*. Our IP Ensemble consistently achieves Pareto improvements across all three datasets, often yielding the best results compared to other methods.

## 7 Defense Against Adaptive Adversaries

In this section, we demonstrate how the influence-based approach can effectively fortify defenses against an adaptive adversary (Tramer et al., 2020; Biggio et al., 2013) that performs evasion attacks on the learning model. In this scenario, the attacker randomly selects a subset of test samples to launch the evasion attack. We defend by proactively trimming the training set by a predetermined amount, although we lack specific knowledge about which samples are adversarial during inference.

In this experiment, we utilize a Logistic Regression model with three datasets: *Bank* (Moro et al., 2014), *CelebA* (Liu et al., 2018), and *JigsawToxicity* (Noever, 2018) to evaluate the defense against adaptive evasion adversaries. The Logistic Regression model is implemented using the `sklearn` library. The model uses L2 regularization with and a maximum iteration limit of 2,048 (Chhabra et al., 2024). The *Bank* dataset consists of features extracted from direct marketing campaigns of a Portuguese banking institution, aiming to predict whether a client will subscribe to a term deposit, including 18,292 training examples, 6,098 validation examples, and 6,098 test examples, with a feature dimension of 51. The *CelebA* dataset is a large-scale face attributes dataset with more than 200,000 celebrity images annotated with 40 binary attributes; we focus on a subset comprising 62,497 training examples, 20,833 validation examples, and 20,833 test examples, each with a feature dimension of 39. The *JigsawToxicity* dataset includes comments from Wikipedia labeled for toxicity prediction, containing 18,000 training examples, 6,000 validation examples, and 6,000 test examples with a feature dimension of 385. Following the protocol in Section 3, we consider a white-box adversary (Megyeri et al., 2019) to craft adversarial samples. For each sample $(x_j, y_j)$ in the validation set $\mathcal{V}$, we perturb it by changing the feature $x'_j = x_j - \gamma \frac{\hat{\theta}^\top x_j + b}{\hat{\theta}^\top \hat{\theta}} \hat{\theta}$ and keeping $y_j$ unchanged. The attacker perturbs between 5%

Table 2: Defense performance of various influence function-based methods under the relabeling and reweighting strategies on *Bank*, *CelebA*, and *JigsawToxicity* datasets

| Defense Strategy | Relabeling | | | | Reweighting | | | |
|---|---|---|---|---|---|---|---|---|
| | *Bank* | *CelebA* | *JigsawToxicity* | Avg | *Bank* | *CelebA* | *JigsawToxicity* | Avg |
| Pre | 80.31 | 85.26 | 73.58 | 79.72 | 80.31 | 85.26 | 73.58 | 79.72 |
| Post | 70.79 ± 3.71 | 75.33 ± 3.04 | 66.18 ± 3.52 | 70.77 | 70.79 ± 3.71 | 75.33 ± 3.04 | 66.18 ± 3.52 | 70.77 |
| LiSSA (Koh & Liang, 2017) | 86.07 ± 0.39 | 68.68 ± 1.39 | **73.30** ± 2.96 | 76.02 | 78.69 ± 3.56 | 73.23 ± 3.70 | 70.13 ± 1.00 | 74.02 |
| EKFAC (Grosse et al., 2023) | 87.38 ± 0.89 | **77.36** ± 1.94 | 70.04 ± 0.96 | 78.26 | 83.50 ± 5.90 | **75.11** ± 1.73 | 70.05 ± 0.76 | 76.22 |
| DataInf (Kwon et al., 2023) | **87.46** ± 2.44 | **77.36** ± 1.85 | 70.39 ± 0.84 | 78.40 | 85.99 ± 2.49 | 74.74 ± 1.68 | 70.82 ± 0.35 | 77.18 |
| Self-TracIn (Thakkar et al., 2023) | 86.12 ± 1.10 | 75.58 ± 2.97 | 68.31 ± 2.73 | 76.67 | 85.22 ± 1.72 | 73.12 ± 3.71 | 71.37 ± 1.81 | 76.57 |
| Self-LiSSA (Bejan et al., 2023) | 78.97 ± 3.40 | 75.55 ± 2.99 | 69.18 ± 1.76 | 74.57 | 85.16 ± 1.74 | 73.17 ± 3.72 | **71.41** ± 1.86 | 76.58 |
| IP (Ours) | 87.45 ± 0.27 | 77.28 ± 1.87 | 70.43 ± 0.87 | 78.38 | 86.17 ± 2.04 | 75.10 ± 1.71 | 70.82 ± 0.36 | 77.36 |
| IP Ensemble (Ours) | 87.45 ± 0.32 | 77.30 ± 1.89 | 70.51 ± 1.00 | **78.42** | **86.44** ± 2.58 | 75.10 ± 1.73 | 70.67 ± 0.27 | **77.40** |

and 25% of the test set samples at random. By quantifying the impact of samples on model robustness, we trim 5% detrimental samples in the training set through influence functions. The boxplot depicted in Figure 4 demonstrates the performance variations across various influence function-based methods over 10 distinct attacks. Since the Logistic Regression model is convex, TDA simplifies to EKFAC, and TracIn and GEX simplify to IP; therefore, we omit their duplicate results in this figure and the following table. For the convex case, the existence of the inverse Hessian matrix is guaranteed, and LiSSA is the most suitable algorithm for influence estimation. Even though, among these influence function-based methods, our IP Ensemble demonstrates competitive efficacy with the best or the second best on these three datasets. These datasets cover a range of tasks from marketing prediction and face attribute recognition to toxicity detection, providing a robust evaluation of the model's defense mechanisms against adaptive adversaries.

In addition to the trimming strategy, we continue to explore relabeling and reweighting strategies to tackle detrimental samples. The relabeling strategy (Kong et al., 2021) changes the identified detrimental samples from their original classes into another class, since the datasets we used here are binary classes, we directly flip their labels. The reweighting strategy (Thakkar et al., 2023) takes the influence score of each sample as the exponential weight with a softmax normalization, and then trains a model with weighted samples. We report the defense performance of various influence function-based methods and our IP Ensemble under the relabeling and reweighting strategies in Table 2. Under the relabeling strategy, DataInf achieves the best results on *Bank* and *CelebA* with 87.46% and 77.36% accuracy, respectively. However, LiSSA is not stable as other methods on *CelebA* with only 68.68%, which is even worse than the performance of post attack. Our IP ensemble method continues to achieve competitive performance, with accuracy scores of 87.45%, 77.30%, and 70.51% on three datasets with the second-best performance among all influence functions-based methods. For the reweighting strategy, the influence function-based methods are still effective on *Bank* and *JigsawToxicity*, which outperform the post-attack by a large margin, but achieve similar performance with post attack on *CelebA*. Notably, when considering the average accuracy of both the relabeling and reweighting strategies, our IP Ensemble method performs the best.

# 8 Conclusion

In this paper, we revisited and simplified the TracIn method into the Inner Product (IP) formulation, which substitutes the inverse of the Hessian matrix with an identity matrix offers a practical and computationally efficient solution to estimating sample influence. Based on that, we extended our IP to measure the sample influence on fairness and robustness. Continually, we enhanced the generalization of IP by introducing an ensemble strategy. We verified the effectiveness of IP on synthetic datasets and extensive evaluations on noisy label detection, data curation for fair NLP model fine-tuning, and defense against adaptive adversarial attacks. Overall, our IP Ensemble highlighted the potential of simple, yet powerful, approximations in influence estimation for practical use.

## Acknowledgment

We gratefully acknowledge the support of the Google TPU Builders Program, which provided support and access to computational resources, and Google Tunix library for post-training functionality and flexibility to enable this work. We also thank the anonymous reviewers and the area chair for their valuable feedback and constructive suggestions.

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

# A    Additional Experimental Results and Analysis

Figure 5: Illustrating our inner product approach on measuring fairness and robustness. **A** and **D** illustrate the same 2D linearly separable synthetic dataset in Figure 1 trained using a Logistic Regression model for binary classification, where the solid and dashed point boundaries denote the majority and minority subgroups. **B** and **E** represent the validation set. **C** and **F** show the estimated influence on fairness and robustness by the traditional influence function and our IP method.

## A.1    IP for Measuring Fairness and Robustness

Similar to Figure 1, we conduct the experiments based on Eqs. (3) and (4) to demonstrate the effectiveness of our IP in measuring the sample influence on fairness and robustness. Figure 5 shows the relationship between IP and the traditional influence function, indicating IP is a good surrogate of the traditional influence function in measuring fairness and robustness.

Table 3: Performance of IP Ensemble with different rates of removed samples on *CIFAR-10N-a*

| Method | Ensemble Size | Remove Rate | ACC |
|---|---|---|---|
| Cross Entropy | 5 | 0 | 91.62 |
| IP Ensemble (Ours) | 5 | 0.025 | 92.29 ± 0.16 |
| IP Ensemble (Ours) | 5 | 0.050 | 92.26 ± 0.19 |
| IP Ensemble (Ours) | 5 | 0.075 | 92.50 ± 0.13 |
| IP Ensemble (Ours) | 5 | 0.100 | 92.37 ± 0.15 |

Table 4: Performance of different dropout rate and ensemble size of IP Ensemble on *CIFAR-10N-a*, *CIFAR-10N-r*, *CIFAR-10N-w*, and *CIFAR-100N*

| Ensemble Size | Dropout Rate | CIFAR-10N-a | CIFAR-10N-r | CIFAR-10N-w | CIFAR-100N |
|---|---|---|---|---|---|
| 5 | 0.01 | $92.26 \pm 0.19$ | $91.28 \pm 0.29$ | $86.50 \pm 0.35$ | $62.25 \pm 0.54$ |
| 5 | 0.1 | $92.26 \pm 0.19$ | $91.28 \pm 0.29$ | $86.50 \pm 0.35$ | $62.25 \pm 0.54$ |
| 5 | 0.5 | $92.26 \pm 0.19$ | $91.28 \pm 0.29$ | $86.50 \pm 0.35$ | $62.25 \pm 0.54$ |
| 1 | 0.01 | $92.42 \pm 0.17$ | $90.82 \pm 0.08$ | $86.31 \pm 0.35$ | $60.59 \pm 0.20$ |
| 5 | 0.01 | $92.26 \pm 0.19$ | $91.28 \pm 0.29$ | $86.50 \pm 0.35$ | $62.25 \pm 0.54$ |
| 10 | 0.01 | $92.58 \pm 0.04$ | $91.32 \pm 0.29$ | $86.89 \pm 0.37$ | $61.82 \pm 0.61$ |
| 15 | 0.01 | $92.27 \pm 0.09$ | $91.27 \pm 0.28$ | $86.47 \pm 0.41$ | $61.59 \pm 0.34$ |
| 20 | 0.01 | $92.41 \pm 0.15$ | $91.33 \pm 0.26$ | $86.65 \pm 0.16$ | $62.25 \pm 0.11$ |

Table 5: Accuracy results of influence-based methods with ViT and MLP-mixer as based models on *CIFAR10N-a*, *CIFAR-100N*, and *Animal-10N* with 5% detrimental samples removed.

| Methods / Datasets | CIFAR-10N-a | CIFAR-100N | Animal-10N | Avg. |
|---|---|---|---|---|
| ViT | 81.87 | 48.53 | 76.20 | 68.87 |
| LiSSA (Koh & Liang, 2017) | $81.83 \pm 0.19$ | $48.47 \pm 0.24$ | $76.90 \pm 0.35$ | 69.07 |
| TracIn (Pruthi et al., 2020) | $81.59 \pm 0.21$ | $48.18 \pm 0.28$ | $76.21 \pm 0.58$ | 68.66 |
| EKFAC (Grosse et al., 2023) | $80.79 \pm 0.56$ | $48.90 \pm 0.30$ | $76.85 \pm 0.30$ | 68.85 |
| DataInf (Kwon et al., 2023) | $81.68 \pm 0.21$ | $49.00 \pm 0.22$ | $76.70 \pm 0.27$ | 69.13 |
| Self-TracIn (Thakkar et al., 2023) | $81.88 \pm 0.12$ | $49.25 \pm 0.21$ | $76.95 \pm 0.25$ | 69.36 |
| Self-LiSSA (Bejan et al., 2023) | $82.89 \pm 0.12$ | $49.30 \pm 0.18$ | $76.80 \pm 0.28$ | 69.66 |
| TDA (Bae et al., 2024) | $81.89 \pm 0.28$ | $49.05 \pm 0.54$ | $76.44 \pm 0.32$ | 69.12 |
| GEX (Kim et al., 2024) | $\mathbf{82.99} \pm 0.24$ | $49.78 \pm 0.20$ | $76.88 \pm 0.29$ | 69.88 |
| IP (Ours) | $81.93 \pm 0.22$ | $49.12 \pm 0.20$ | $76.80 \pm 0.22$ | 69.28 |
| IP Ensemble (Ours) | $81.96 \pm 0.13$ | $\mathbf{51.01} \pm 0.17$ | $\mathbf{77.20} \pm 0.24$ | **70.06** |
| MLP-Mixer | 74.42 | 35.63 | 72.69 | 60.91 |
| LiSSA (Koh & Liang, 2017) | $75.35 \pm 0.47$ | $36.50 \pm 0.21$ | $72.89 \pm 0.28$ | 61.58 |
| TracIn (Pruthi et al., 2020) | $74.43 \pm 0.28$ | $35.21 \pm 0.35$ | $72.71 \pm 0.43$ | 60.78 |
| EKFAC (Grosse et al., 2023) | $75.49 \pm 0.28$ | $36.34 \pm 0.24$ | $73.12 \pm 0.19$ | 61.65 |
| DataInf (Kwon et al., 2023) | $75.10 \pm 0.45$ | $35.78 \pm 0.29$ | $73.17 \pm 0.21$ | 61.35 |
| Self-TracIn (Thakkar et al., 2023) | $75.88 \pm 0.21$ | $36.72 \pm 0.19$ | $73.81 \pm 0.54$ | 62.14 |
| Self-LiSSA (Bejan et al., 2023) | $75.41 \pm 0.38$ | $36.05 \pm 0.27$ | $72.56 \pm 0.34$ | 61.34 |
| TDA (Bae et al., 2024) | $74.89 \pm 0.31$ | $36.84 \pm 0.38$ | $71.88 \pm 0.43$ | 61.20 |
| GEX (Kim et al., 2024) | $75.57 \pm 0.27$ | $36.85 \pm 0.25$ | $\mathbf{73.89} \pm 0.43$ | 62.10 |
| IP (Ours) | $75.04 \pm 0.23$ | $36.15 \pm 0.18$ | $73.47 \pm 0.20$ | 61.55 |
| IP Ensemble (Ours) | $\mathbf{75.77} \pm 0.19$ | $\mathbf{37.12} \pm 0.18$ | $73.80 \pm 0.11$ | **62.23** |

### A.2 Performance of IP Ensemble with different rates of removed samples

In our paper, we explore different rates of removed samples on *CIFAR-10N-a*. Table 3 shows the performance of IP Ensemble with different rates of removed samples. Except for not removing, there is no significant difference in the remaining results.

### A.3 Parameter Analysis on Dropout Rate and Ensemble Size

Table 4 shows the performance of different dropout rates and ensemble sizes of IP Ensemble. As can be observed, our IP Ensemble is not sensitive to dropout rate, and its performance increases with a large ensemble size, demonstrating the effectiveness of the ensemble strategy. It is also worth noting that although our IP Ensemble runs fast, it might take a longer time to calculate the sample gradient.

### A.4 Performance on ViT and MLP-mixer

To verify the effectiveness of our IP ensemble across different network architectures, we conducte experiments using ViT (Dosovitskiy, 2020) and MLP-Mixer (Tolstikhin et al., 2021) on various vision datasets in Table 5. We use a batch size of 512 for all experiments in this section. The models are trained on *CIFAR-10N* and *CIFAR-100N* datasets for 100 epochs, and on *Animal-10N* for 400 epochs. The learning rate was set to $1 \times 10^{-4}$ for ViT and $1 \times 10^{-3}$ for MLP-Mixer.

Our IP Ensemble consistently demonstrates superior performance compared to the vanilla models and other baseline methods on different base models. Specifically, when trained on ViT, IP Ensemble achieves an average accuracy of 70.06%, outperforming the vanilla ViT's average accuracy of 68.87%. This improvement is observed across all datasets, with notable gains on *CIFAR-100N* (51.01% versus 48.53%) and *Animal-10N* (77.20% versus 76.20%). Additionally, compared to baseline methods, our IP Ensemble achieves the best performance with ViT or MLP-Mixer based models, indicating the effectiveness of our methods on different base models.

### A.5 Time Complexity and Execution Time of Various Influence Function-based Methods on Vision Datasets

Table 6: Computational complexity and runtime (seconds) of influence-function-based methods ($n$ is #training samples and $p$ is #model parameters, $k$ is #checkpoints or #ensemble size with $k=1$ here. "-" denotes no runs.)

| Method | Type | Time Complexity | CIFAR-10N | CIFAR-100N | Animal-10N |
|---|---|---|---|---|---|
| Exact by Eq. (1) | Hessian-based | $\mathcal{O}(np^3)$ | - | - | - |
| LiSSA (Koh & Liang, 2017) | Hessian-based | $\mathcal{O}(np)$ | 7.67 | 34.59 | 7.46 |
| TracIn (Pruthi et al., 2020) | Hessian-free | $\mathcal{O}(npk)$ | 0.03 | 0.28 | 0.03 |
| EKFAC (Grosse et al., 2023) | Hessian-based | $\mathcal{O}(np^2)$ | 22.54 | 192.58 | 23.89 |
| DataInf (Kwon et al., 2023) | Hessian-based | $\mathcal{O}(np)$ | 11.50 | 45.29 | 10.89 |
| Self-TracIn (Thakkar et al., 2023) | Self-influence | $\mathcal{O}(npk)$ | 0.15 | 1.41 | 0.15 |
| Self-LiSSA (Bejan et al., 2023) | Self-influence | $\mathcal{O}(np)$ | 14.57 | 54.39 | 14.40 |
| GEX (Kim et al., 2024) | Hessian-free | $\mathcal{O}(npk)$ | 0.03 | 0.28 | 0.03 |
| TDA (Bae et al., 2024) | Hessian-based | $\mathcal{O}(np^2k)$ | 23.98 | 193.79 | 24.77 |
| IP Ensemble (Ours) | Hessian-free | $\mathcal{O}(npk)$ | 0.03 | 0.28 | 0.03 |

Table 6 shows the time complexity of various influence function-based methods. Except for the vanilla calculation, all other methods have linear time complexity in terms of the sample size. However, they have large divergence in real execution time.

We omit the specific timing details for the sample gradient, which are readily available during the base model's training phase. Besides, utilizing vmap in Pytorch, we can efficiently compute gradients in parallel. For example, in Section 5, it only takes 61 seconds and 4 seconds to calculate $\nabla v^{\text{util}}$ and $\nabla_{\hat{\theta}} \ell(z_i; \hat{\theta})$ on *CIFAR-10N* dataset, respectively.

