# OpenReview forum: "Revisit, Extend, and Enhance Hessian-Free Influence Functions"
_TMLR — Accepted by TMLR_

### Review · Reviewer_jUHb · 2025-12-23

**Summary Of Contributions:**

This paper studies data-centric learning and considers a Hessian-free approach to influence functions. In the estimation of a training sample's influence, the inverse Hessian matrix is replaced with an identity matrix. The resulting gradient "Inner Product" (IP) gives a simple similarity measure that can be more effective and scalable than computationally expensive methods for deep learning models. The authors also extend this method to improve model fairness and adversarial robustness.

**Additional Comments:**

### Section 2
- Page 3, first para - you mean "fairness 'in' ML" right?

### Section 3
- Why -ve sign in $z_i$ in (1)?
- Page 4, 2nd para, last sentence - do you mean 'exceed the "disadvantages" of simply'? Omitting the Hessian cannot have advantages - at best the resulting errors are manageable.
- In (3), why is there $\hat{\theta}$ in the subscript of the first gradient? Also, the order of model and data in $f$ and $\ell$ are reverse. Can you make it consistent?
- For IP, as you state in para 2 of page 4, "A high IP score indicates that the target training sample is similar to the validation set". However, what is the intuition for IP in (4)? If gradient of a sample $z_i$ is aligned with the gradients at the perturbed samples, it is good because then you can move in the direction $-\nabla \ell(z_i; \hat{\theta})$ and increase robustness - is that the reasoning?

### Section 4
- Can make it more concise. E.g., several sentences in the first para are repeated.

### Section 5
- Clarification question: in the 5 retraining iterations, do you look for 5% detrimental samples from the full training dataset each time? Or does the dataset keeps shrinking each time by 5% further after each retraining? Or is it the case that you remove detrimental samples only once, and retrain with the remaining 95% dataset for 5 independent runs? Also, it's not clear from caption of Table 1 that 5 retraining iterations happen.
- Page 8, line 2: what are the "different levels" of detrimental samples?
- Table 1: for all the datasets, the performance of IP Ensemble has overlapping confidence intervals with some other method. Does it have any significant utility then? Also, I don't see the benefit of the last column. Maybe averaging CIFAR-10N's across different noise levels would still make sense.
- In Table 6, what are the units? Seconds?

### Section 6
- How does IP (without ensemble) perform in Fig 3?
- When removing the detrimental samples according to both utility and fairness, do you remove the 5% worst samples for utility AND 5% worse for fairness? Or some other way? Please clarify.

### Section 7
- The dashed brown line in Fig 4 - what is the average over: the 10 attacks? And why is LiSSA so much worse on CelebA with relabeling?

**Audience:**

Yes

**Audience Explanation:**

Yes, as suggested by the numerous papers from recent ML papers across venues.

**Claims And Evidence:**

Yes

**Claims Explanation:**

Yes, the authors do extensive experiments to back their claims. I do have some comments/questions on them - see below.

**Requested Changes:**

### Section 2
- In the "Miscellaneous" para, the discussion seems very unrelated across sentences. How are these different works related to the work in this paper? And would making them chronological help?

### Section 7
- Since the attacker is sometimes attacking more samples than those removed by influence functions, there should come a point at which these methods will do worse than pre-attack. Can you show some results about the levels of attacks when different methods become worse than "pre"?

### Generic
- I would love to see some experiments where IP does significantly worse than Hessian-based approaches, to highlight some limitations of when IP doesn't work.
- See more comments below.

---

> ### Author Response · Authors · 2026-02-19
> **Response to Reviewer jUHb (1/5)**
>
> Thank you, Reviewer jUHb, for your time and effort on reviewing our paper. From the bottom of our hearts, your comments are highly constructive to improve the quality of our paper, which we are happy to follow. Truly appreciated. In the follows, we elaborate how to modify our paper according to your comments. In the revised manuscript, all changes (including additions and modifications) are highlighted in blue.
>
> **Related Work**. We fully agree that the miscellaneous part in the section of related work is somehow unrelated across sentences. We aim to provide the open debate on influence function in the deep scenarios in miscellaneous part, which we would like to provide a paramount view to reader. For the modification plan, we would like to follow Reviewer jUHb's suggest and reorganize this part in the chronological order.
>
>
> **Different Attack Levels**. We thank the reviewer for the suggestion. In our original Section 7, the attacker perturbs a random fraction of test samples (5--25%) while the defender applies a fixed 5% training-set budget in relabeling and reweighting. To explicitly identify at which attack levels each defense becomes worse than the clean pre-attack baseline, we additionally sweep the attack ratio over
>
> $$
> \{5, 10, 15, 20, 25, 30, 40, 50\}\%
> $$
>
> (i.e., extending beyond the original 25%). For each ratio, we repeat the random choice of attacked test points 5 times and report the attacked test accuracy (mean ± std). Table 1 summarizes the earliest attack ratio at which each method drops below pre-attack baseline.
>
> The crossing behavior depends on both the dataset and the defense strategy. On *CelebA*, all methods fall below Pre at a 5% attack ratio under both relabeling and reweighting. On Bank, several methods cross under reweighting at 5%, while under relabeling LiSSA, IP, IP Ensemble, and DataInf remain above Pre up to 50% (reported as >50% in Table 1). Our attacks are generated on the pre-attack model and evaluated on the defended model (transfer setting); fully adaptive attacks or stronger perturbations are expected to eventually induce crossings. Table 2 and Table 3 report the full sweeps.
>
> | Strategy | Dataset | LiSSA | IP | IP Ensemble | DataInf | Self-LiSSA |
> |---|---|---:|---:|---:|---:|---:|
> | relabel | *Bank* | >50% | >50% | >50% | >50% | 15% |
> | relabel | *CelebA* | 5% | 5% | 5% | 5% | 5% |
> | relabel | *JigsawToxicity* | 10% | 5% | 5% | 5% | 5% |
> | reweight | *Bank* | 5% | 5% | 5% | 5% | 15% |
> | reweight | *CelebA* | 5% | 5% | 5% | 5% | 5% |
> | reweight | *JigsawToxicity* | 5% | 5% | 5% | 5% | 5% |
>
> *Table 1. Earliest attack ratio where the defense accuracy becomes worse than the clean Pre accuracy (computed from Tables 2--3). We report >50% when no crossing is observed up to 50% attack ratio.*

---

> > ### Author Response · Authors · 2026-02-19
> > **Response to Reviewer jUHb (2/5)**
> >
> > *Table 2. Attack-level sweep results under **relabel** defense. Accuracy (%) on the attacked test set (mean ± std over 5 runs). Pre is the clean (non-attacked) accuracy of the base model.*
> >
> > | Dataset | Attack ratio | Pre | Post | LiSSA | IP | IP Ensemble | DataInf | Self-LiSSA |
> > |---|---:|---:|---:|---:|---:|---:|---:|---:|
> > | *Bank* | 5% | 80.31 | 77.29 ± 0.12 | 85.39 ± 0.43 | 84.40 ± 2.19 | 83.58 ± 1.61 | 84.67 ± 2.09 | 84.78 ± 0.13 |
> > | *Bank* | 10% | 80.31 | 74.31 ± 0.39 | 85.98 ± 0.20 | 87.77 ± 0.52 | 87.64 ± 0.59 | 87.91 ± 0.35 | 82.06 ± 0.39 |
> > | *Bank* | 15% | 80.31 | 71.21 ± 0.24 | 86.33 ± 0.50 | 87.68 ± 0.18 | 87.67 ± 0.13 | 87.66 ± 0.19 | 79.40 ± 0.11 |
> > | *Bank* | 20% | 80.31 | 68.25 ± 0.35 | 86.06 ± 0.32 | 87.55 ± 0.30 | 87.60 ± 0.34 | 87.61 ± 0.34 | 76.91 ± 0.38 |
> > | *Bank* | 25% | 80.31 | 65.22 ± 0.65 | 86.06 ± 0.28 | 87.32 ± 0.29 | 87.34 ± 0.35 | 87.41 ± 0.30 | 74.03 ± 0.31 |
> > | *Bank* | 30% | 80.31 | 62.34 ± 0.49 | 85.82 ± 0.14 | 87.25 ± 0.30 | 87.29 ± 0.29 | 87.42 ± 0.25 | 71.24 ± 0.44 |
> > | *Bank* | 40% | 80.31 | 56.37 ± 0.39 | 85.32 ± 0.17 | 87.19 ± 0.25 | 87.32 ± 0.22 | 87.45 ± 0.27 | 65.76 ± 0.38 |
> > | *Bank* | 50% | 80.31 | 50.07 ± 0.78 | 84.85 ± 0.29 | 86.87 ± 0.17 | 86.86 ± 0.17 | 87.07 ± 0.21 | 60.24 ± 0.65 |
> > | *CelebA* | 5% | 85.26 | 81.71 ± 0.09 | 72.43 ± 0.18 | 81.53 ± 0.11 | 81.50 ± 0.20 | 81.54 ± 0.11 | 81.87 ± 0.07 |
> > | *CelebA* | 10% | 85.26 | 78.07 ± 0.11 | 69.84 ± 0.17 | 78.92 ± 0.10 | 78.96 ± 0.14 | 78.99 ± 0.12 | 78.31 ± 0.11 |
> > | *CelebA* | 15% | 85.26 | 74.65 ± 0.13 | 68.13 ± 0.18 | 76.87 ± 0.14 | 76.86 ± 0.14 | 76.97 ± 0.15 | 74.91 ± 0.13 |
> > | *CelebA* | 20% | 85.26 | 71.28 ± 0.22 | 67.01 ± 0.15 | 74.87 ± 0.13 | 74.87 ± 0.13 | 74.97 ± 0.11 | 71.53 ± 0.20 |
> > | *CelebA* | 25% | 85.26 | 67.65 ± 0.27 | 65.94 ± 0.13 | 72.75 ± 0.15 | 72.73 ± 0.18 | 72.87 ± 0.16 | 67.89 ± 0.21 |
> > | *CelebA* | 30% | 85.26 | 63.98 ± 0.23 | 64.75 ± 0.32 | 70.87 ± 0.28 | 70.85 ± 0.23 | 71.08 ± 0.23 | 64.31 ± 0.27 |
> > | *CelebA* | 40% | 85.26 | 56.96 ± 0.32 | 62.81 ± 0.08 | 66.78 ± 0.10 | 66.77 ± 0.15 | 67.05 ± 0.11 | 57.35 ± 0.33 |
> > | *CelebA* | 50% | 85.26 | 49.99 ± 0.30 | 60.77 ± 0.14 | 62.81 ± 0.21 | 62.80 ± 0.16 | 63.17 ± 0.20 | 50.44 ± 0.29 |
> > | *JigsawToxicity* | 5% | 73.58 | 71.28 ± 0.06 | 73.74 ± 0.14 | 71.55 ± 0.13 | 71.76 ± 0.13 | 71.49 ± 0.10 | 71.69 ± 0.10 |
> > | *JigsawToxicity* | 10% | 73.58 | 68.66 ± 0.49 | 73.44 ± 0.21 | 70.98 ± 0.48 | 70.79 ± 0.29 | 70.96 ± 0.48 | 70.34 ± 0.33 |
> > | *JigsawToxicity* | 15% | 73.58 | 66.34 ± 0.32 | 73.28 ± 0.06 | 70.55 ± 0.24 | 70.45 ± 0.35 | 70.59 ± 0.25 | 69.00 ± 0.29 |
> > | *JigsawToxicity* | 20% | 73.58 | 63.75 ± 0.53 | 73.23 ± 0.13 | 69.73 ± 0.42 | 69.86 ± 0.46 | 69.85 ± 0.42 | 68.16 ± 0.25 |
> > | *JigsawToxicity* | 25% | 73.58 | 61.65 ± 0.41 | 73.21 ± 0.16 | 69.37 ± 0.46 | 69.40 ± 0.65 | 69.42 ± 0.48 | 67.07 ± 0.27 |
> > | *JigsawToxicity* | 30% | 73.58 | 59.77 ± 0.51 | 73.25 ± 0.20 | 68.65 ± 0.40 | 68.87 ± 0.31 | 68.66 ± 0.37 | 65.95 ± 0.36 |
> > | *JigsawToxicity* | 40% | 73.58 | 54.57 ± 0.28 | 73.32 ± 0.22 | 66.37 ± 0.38 | 66.49 ± 0.55 | 66.38 ± 0.38 | 63.65 ± 0.20 |
> > | *JigsawToxicity* | 50% | 73.58 | 49.84 ± 0.63 | 73.36 ± 0.07 | 65.13 ± 0.63 | 64.91 ± 0.98 | 65.17 ± 0.75 | 60.96 ± 0.54 |

---

> ### Author Response · Authors · 2026-02-19
> **Response to Reviewer jUHb (3/5)**
>
> *Table 3. Attack-level sweep results under **reweight** defense. Accuracy (%) on the attacked test set (mean ± std over 5 runs). Pre is the clean (non-attacked) accuracy of the base model.*
>
> | Dataset | Attack ratio | Pre | Post | LiSSA | IP | IP Ensemble | DataInf | Self-LiSSA |
> |---|---:|---:|---:|---:|---:|---:|---:|---:|
> | *Bank* | 5% | 80.31 | 77.31 ± 0.18 | 65.51 ± 0.99 | 72.91 ± 4.76 | 74.86 ± 7.54 | 70.31 ± 5.16 | 84.94 ± 0.17 |
> | *Bank* | 10% | 80.31 | 74.33 ± 0.10 | 76.64 ± 2.52 | 86.62 ± 0.56 | 86.91 ± 0.48 | 86.12 ± 0.98 | 82.19 ± 0.05 |
> | *Bank* | 15% | 80.31 | 71.46 ± 0.36 | 79.88 ± 2.20 | 86.99 ± 0.32 | 86.90 ± 0.46 | 87.00 ± 0.39 | 79.55 ± 0.38 |
> | *Bank* | 20% | 80.31 | 68.00 ± 0.25 | 80.94 ± 0.90 | 86.86 ± 0.19 | 86.83 ± 0.16 | 86.93 ± 0.20 | 76.51 ± 0.24 |
> | *Bank* | 25% | 80.31 | 65.05 ± 0.23 | 82.57 ± 0.59 | 86.95 ± 0.12 | 86.92 ± 0.15 | 87.03 ± 0.14 | 73.98 ± 0.20 |
> | *Bank* | 30% | 80.31 | 61.84 ± 0.31 | 81.89 ± 1.05 | 86.45 ± 0.12 | 86.43 ± 0.16 | 86.56 ± 0.13 | 71.04 ± 0.29 |
> | *Bank* | 40% | 80.31 | 56.04 ± 0.35 | 82.59 ± 1.00 | 86.01 ± 0.27 | 86.06 ± 0.29 | 86.11 ± 0.28 | 65.82 ± 0.40 |
> | *Bank* | 50% | 80.31 | 50.09 ± 0.47 | 82.43 ± 0.29 | 85.57 ± 0.24 | 85.56 ± 0.25 | 85.69 ± 0.32 | 60.31 ± 0.16 |
> | *CelebA* | 5% | 85.26 | 81.77 ± 0.07 | 64.69 ± 0.18 | 80.50 ± 0.06 | 80.51 ± 0.16 | 80.48 ± 0.06 | 81.93 ± 0.08 |
> | *CelebA* | 10% | 85.26 | 78.26 ± 0.11 | 65.39 ± 0.07 | 77.61 ± 0.19 | 77.65 ± 0.19 | 77.27 ± 0.18 | 78.43 ± 0.12 |
> | *CelebA* | 15% | 85.26 | 74.82 ± 0.16 | 64.58 ± 0.18 | 76.03 ± 0.09 | 76.08 ± 0.13 | 75.67 ± 0.11 | 75.02 ± 0.10 |
> | *CelebA* | 20% | 85.26 | 71.11 ± 0.15 | 63.50 ± 0.13 | 74.22 ± 0.11 | 74.27 ± 0.09 | 73.81 ± 0.19 | 71.39 ± 0.12 |
> | *CelebA* | 25% | 85.26 | 67.49 ± 0.16 | 62.22 ± 0.21 | 72.38 ± 0.11 | 72.34 ± 0.13 | 72.08 ± 0.11 | 67.77 ± 0.16 |
> | *CelebA* | 30% | 85.26 | 64.01 ± 0.16 | 61.68 ± 0.45 | 70.90 ± 0.21 | 70.92 ± 0.16 | 70.75 ± 0.22 | 64.34 ± 0.18 |
> | *CelebA* | 40% | 85.26 | 56.97 ± 0.16 | 60.91 ± 0.14 | 67.94 ± 0.12 | 67.93 ± 0.10 | 67.94 ± 0.12 | 57.36 ± 0.11 |
> | *CelebA* | 50% | 85.26 | 50.06 ± 0.11 | 60.69 ± 0.08 | 64.90 ± 0.17 | 64.94 ± 0.20 | 65.14 ± 0.18 | 50.54 ± 0.10 |
> | *JigsawToxicity* | 5% | 73.58 | 71.25 ± 0.14 | 71.61 ± 0.14 | 70.27 ± 0.20 | 70.00 ± 0.24 | 70.26 ± 0.22 | 71.24 ± 0.05 |
> | *JigsawToxicity* | 10% | 73.58 | 68.71 ± 0.21 | 71.33 ± 0.21 | 70.67 ± 0.32 | 70.38 ± 0.33 | 70.65 ± 0.32 | 70.20 ± 0.18 |
> | *JigsawToxicity* | 15% | 73.58 | 66.45 ± 0.13 | 71.02 ± 0.09 | 71.19 ± 0.36 | 71.06 ± 0.37 | 71.16 ± 0.36 | 69.57 ± 0.23 |
> | *JigsawToxicity* | 20% | 73.58 | 63.92 ± 0.40 | 70.98 ± 0.27 | 70.82 ± 0.30 | 70.79 ± 0.14 | 70.83 ± 0.28 | 68.80 ± 0.35 |
> | *JigsawToxicity* | 25% | 73.58 | 62.02 ± 0.31 | 70.96 ± 0.19 | 70.84 ± 0.40 | 70.47 ± 0.38 | 70.84 ± 0.45 | 68.05 ± 0.08 |
> | *JigsawToxicity* | 30% | 73.58 | 59.63 ± 0.44 | 70.61 ± 0.34 | 70.22 ± 0.19 | 70.12 ± 0.33 | 70.28 ± 0.21 | 67.02 ± 0.21 |
> | *JigsawToxicity* | 40% | 73.58 | 54.88 ± 0.69 | 71.02 ± 0.22 | 69.47 ± 0.30 | 69.44 ± 0.54 | 69.52 ± 0.30 | 65.56 ± 0.47 |
> | *JigsawToxicity* | 50% | 73.58 | 49.95 ± 0.21 | 71.45 ± 0.10 | 68.41 ± 0.39 | 68.38 ± 0.39 | 68.43 ± 0.42 | 63.40 ± 0.38 |

---

> ### Author Response · Authors · 2026-02-19
> **Response to Reviewer jUHb (4/5)**
>
> **Algorithmic Condition**. Excellent suggestion! All influence function-based methods aim to approximate the value in Eq. (1). Several assumptions (convexity, stationary point) should meet to guarantee the value in Eq. (1) is accurate. That means, if these conditions are well satisfied, high performance is expected. However, there exist errors in practices including the error of non-obeying assumption, the error of approximating the inverse of Hessian matrix, and so on.
>
> Back to Reviewer jUHB's question of when IP does not work, IP replaces the inverse Hessian in Eq. (1) with the identity matrix, so it mainly relies on the gradient-alignment signal. Therefore, for simple settings where the model is relatively convex/strongly convex and the data are low-dimensional, IP may underperform Hessian-inverse-based estimators that can better exploit curvature information. Empirically, we observe such a case in the paper’s Table 2: on *JigsawToxicity* under the relabeling defense, LiSSA achieves $73.30 \pm 2.96$, while IP and IP Ensemble obtain $70.43 \pm 0.87$ and $70.51 \pm 1.00$, respectively.
>
> **Fairness**. Fairness in machine learning refers to the principle that a model’s decisions or predictions should not systematically disadvantage individuals or groups, especially those defined by demographic attributes such as race, gender, age, or socioeconomic status. In Section 6, we use fair score to measure the change of model output by perturbing the input demographic information.
>
> **Eq. (1)**. Eq. (1) is derived by removing the sample $z_i$. Usually, “-” is widely used to denote an element excluded from a set.
>
> **Exceed Disadvantages**. Thank you for the question, and sorry for any confusion. Our goal in discussing the non-convex setting is to clarify why inner-product-based (IP) influence avoids key limitations of Hessian-inverse-based methods. In non-convex models, the Hessian inverse may not exist, making such approximation methods both computationally expensive and theoretically ill-posed. As a result, the approximation errors are difficult to quantify, and there is no guarantee that the identified samples are truly beneficial or detrimental.
>
> In contrast, IP-based influence does not rely on Hessian inversion or the assumptions of influence functions. As long as the model is reasonably well trained, adding validation-like samples to the training set must reduce the validation loss. Therefore, while IP may not always identify the most influential samples, it provides a reliable guarantee on the polarity of influence, which is particularly desirable in non-convex settings. We have modified this paragraph and made this point more clear.
>
> **Typos in Eq. (3)**. Thank you for pointing this out. We have revised it in the new version. We use $\hat{\theta}$ in the subscript of the first gradient to demonstrate the gradient is obtained under the model parameter $\hat{\theta}$.
>
> **Eq. (4)**. Your understanding is correct! Note that the validation set in Eq. (4) is the perturbed samples. Therefore, moving the model parameters in the direction could increase robustness.
>
> **Section 4**. Good suggestion. We have revised Section 4 to be shorter and more concise by removing redundant descriptions of the experimental setup.
>
> **Clarification Questions**. We apologize for the lack of clarity. We have clarified these points in the revised version.
> (i) In our experiments, for each run, we first identify the bottom 5% most detrimental samples using each algorithm, then retrain the model on the remaining 95% of the data. This procedure is repeated five times, and the results reported in Tables 1 and 5 are averaged over these five runs.
> (ii) The detrimental samples at different levels mean the aggregate, random, and worst noises in *CIFAR-10* dataset.
> (iii) In general, ensemble methods are expected more robust than the individual method, sometimes having slight performance improvement, but using more computational resources. Here our IP ensemble utilizes the dropout mechanism, which makes the computational cost affordable. The last column reports the average performance across different noise types and datasets, which provides a concise summary of each method’s overall robustness and consistency across settings, instead of relying on gains from a single noise type or dataset.
> (iv) The unit of Table 6 is second.
> (v) We jointly account for utility and fairness by aggregating their gradient-based influence scores into a single ranking and remove the bottom 5% samples under this joint criterion, rather than taking a union or intersection.

---

> ### Author Response · Authors · 2026-02-19
> **Response to Reviewer jUHb (5/5)**
>
> **IP in Figure 3**. We report IP without ensembling as the row IP in Table 4. Compared to the baseline, IP is:
> (i) *RTE*: higher accuracy and better fairness (68.95 vs. 67.99 Acc; 0.61 vs. 1.83 Fair);
> (ii) *CoLA*: higher accuracy but worse fairness (62.54 vs. 58.69; 5.49 vs. 3.66);
> (iii) *QNLI*: higher accuracy with slightly worse fairness (92.57 vs. 91.05; 0.33 vs. 0.31).
>
> *Table. Accuracy and fair score on* *RTE*/*CoLA*/*QNLI* *(Fig. 3). CoLA “Acc” is Matthews correlation × 100. Lower fair score is better.*
>
> | Method | RTE Acc ↑ | RTE Fair ↓ | CoLA Acc ↑ | CoLA Fair ↓ | QNLI Acc ↑ | QNLI Fair ↓ |
> |---|---:|---:|---:|---:|---:|---:|
> | Baseline | 67.99 | 1.83 | 58.69 | 3.66 | 91.05 | 0.31 |
> | TracIn | 64.87 | 1.22 | 59.05 | 2.28 | 92.63 | 0.36 |
> | LiSSA | 64.94 | 0.61 | 60.31 | 3.43 | 92.51 | 0.41 |
> | EKFAC | 67.99 | 1.22 | 58.37 | 3.43 | 91.65 | 0.36 |
> | DataInf | 69.82 | 0.61 | 59.30 | 5.03 | 92.63 | 0.31 |
> | Self-TracIn | 68.29 | 0.00 | 59.30 | 3.89 | 91.99 | 0.27 |
> | Self-LiSSA | 68.29 | 0.00 | 57.11 | 4.35 | 92.63 | 0.45 |
> | GEX | 68.90 | 1.83 | 53.54 | 2.51 | 92.65 | 0.35 |
> | TDA | 66.46 | 1.83 | 52.40 | 3.43 | 92.97 | 0.39 |
> | IP | 68.95 | 0.61 | 62.54 | 5.49 | 92.57 | 0.33 |
> | IP Ensemble | 69.82 | 0.00 | 60.82 | 2.51 | 92.51 | 0.20 |
>
> **Figure 4**.
> The dashed brown line represents the pre-attack (clean) accuracy, serving as a reference baseline. It is averaged over 10 independent attack iterations, where each iteration randomly perturbs 5%–25% of test samples.
>
> LiSSA is a Hessian-based influence method that relies on approximating the effect of the true inverse Hessian. In practice, such approximations become less accurate and less stable when the Hessian is poorly conditioned, especially in high-dimensional models. In our defense experiments, *CelebA* is much higher-dimensional than the tabular *Bank* dataset and empirically shows much less stable behavior. This is consistent with Hessian-based approximations being more error-prone under poor conditioning. Concretely, in Table 2 the accuracy of LiSSA under the relabeling defense drops to $68.68 \pm 1.39$ on *CelebA*, while it remains $86.07 \pm 0.39$ on *Bank*.
>
> The choice of defense also affects how sensitive the results are to influence estimation errors. Relabeling is more sensitive than removal or reweighting because it makes hard decisions about which labels to flip. Even mild mis-ranking of influence scores can cause the defense to flip benign samples (or miss truly detrimental ones), which can lead to a large drop in accuracy. By contrast, removal or reweighting does not make discrete label changes and averages the effect over many samples, so it is more robust to moderate noise in the influence scores. The same pattern appears in our fairness-aware fine-tuning results in Figure 3: Hessian-inverse-based methods (including LiSSA and Self-LiSSA) more often show utility–fairness trade-offs or even Pareto deterioration on some tasks, suggesting less stable sample ranking in these more complex, high-dimensional settings.

---

### Review · Reviewer_DQj1 · 2026-02-02

**Summary Of Contributions:**

This work investigates the use of influence functions to enhance machine learning model performance by refining training data quality. In contrast to standard methods that rely on the Hessian matrix, the author proposes a new approach that replaces the Hessian with an identity matrix. This modification significantly reduces the computational complexity in high-dimensional spaces. The author provides both simulation results and real-world experiments to demonstrate the efficiency of this method.

**Audience:**

Yes

**Audience Explanation:**

The paper provides a computationally efficient alternative to the Hessian bottleneck in influence functions, offering a scalable heuristic that is highly relevant to TMLR’s audience in interpretability and data valuation.

**Claims And Evidence:**

Yes

**Claims Explanation:**

The author provides detailed experimental results to demonstrate the efficiency of the inner-product method across several domains, including fairness and robustness, including noisy label detection, sample selection for large language model (LLM) fine-tuning, and defense against adversarial attacks.

**Requested Changes:**

1: Section 4 contains redundant descriptions of the experimental setup. Specifically, the descriptions of the Logistic Regression model on linear data and the three-layer MLP on the half-moon dataset appear multiple times.

2: The synthetic experiments in Figure 1 lack realism because the validation set is noise-free. To better simulate real-world conditions, the author should evaluate the method using a validation set with a subset of flipped labels (e.g., 10%). Furthermore, including a plot that utilizes the actual Hessian matrix—rather than the identity matrix approximation—is essential to visualize and quantify the approximation error.

3: For the real-world experiments, the inclusion of a baseline using the exact Hessian-based influence function is highly recommended. Comparing the proposed Inner-Product (IP) method against the standard Hessian method would clarify which performance discrepancies arise from the IP approximation itself versus the inherent limitations of influence functions.

---

> ### Author Response · Authors · 2026-02-19
> **Response to Reviewer DQj1**
>
> We thank the reviewer for the thoughtful comments and constructive suggestions.
>
> **Redundant descriptions**.
> Thank you for pointing this out. We agree that Section 4 contained redundant descriptions, and we have removed the repeated passages in the revised version. All related changes are highlighted in blue.
>
> **Flip 10% labels**.
> We would like to clarify the role of the validation set. In standard practice, the validation set is primarily used for model selection (e.g., hyperparameter tuning and early stopping) and is therefore assumed to be as clean and trusted as possible.
>
> Introducing label noise into the validation set changes the setting because it corrupts the selection signal itself. That said, we agree that this is still a useful stress test to evaluate robustness when the available reference set is noisy. We follow Reviewer DQj1's suggestion and conducted additional experiments with a noisy validation set where 10% of labels are randomly flipped to simulate real-world labeling noise. The resulting plots are available in our anonymous repository (see https://anonymous.4open.science/r/IP_fig-1A36/figure1_noisy.pdf), where Figure C and Figure F refer to the corresponding subfigures in that page.
>
> In Figure C (Linear case), after introducing validation label noise, both the Hessian-inverse influence estimates (x-axis) and the inner-product (IP) scores (y-axis) exhibit limited separation between outliers (marked as X) and normal training samples, i.e., both signals become less reliable for identifying detrimental samples in this setting.
>
> In contrast, in Figure F (Non-Linear case), IP still provides a much clearer separation, where the outliers (X markers) are consistently separated from normal points along the y-axis, while the influence values remain mixed with normal samples along the x-axis. This suggests that, under noisy reference signals, IP can retain better sample discrimination in the non-linear setting compared to Hessian-inverse-based influence estimation.
>
> **Actual Hessian matrix**.
> We appreciate this question, as it allows us to clarify how existing Hessian-based methods work in practice. No existing method computes the exact Hessian inverse for neural networks due to the substantial cost, which requires $\mathcal{O}(d^3)$ and $\mathcal{O}(d^2)$ for time and space complexities with $d$ representing the number of model parameters. LiSSA uses truncated Neumann series; DataInf uses closed-form gradient-based approximations; EKFAC uses Kronecker-factored Fisher approximations. Crucially, these methods approximate the Gauss-Newton matrix, which is positive semi-definite, rather than the true Hessian.
>
> We computed the exact Hessian for the MLP on half-moons using second-order automatic differentiation. We initially attempted to use this exact Hessian to plot an “actual-Hessian” influence and directly visualize the approximation error of replacing $H^{-1}$ with the identity. However, we found that even in this toy non-linear setting the exact Hessian is already indefinite and numerically fragile, making “exact-Hessian inversion” an ill-posed and unstable baseline.
>
> The minimum eigenvalue is $\lambda_{\min} = -75.85$, confirming the Hessian is indefinite. This observation also suggests that, in such non-convex regimes, spending effort to approximate (and then invert) the true Hessian is arguably not meaningful: the matrix is indefinite and highly sensitive to noise, so small approximation errors can lead to qualitatively different influence estimates.
>
> Motivated by this, our method intentionally drops the Hessian term and instead uses the inner product between training and validation gradients. This can be interpreted as a direct measure of similarity (alignment) between how a training sample and the validation set would update the model, which is a more intuitive and task-aligned signal for identifying helpful versus harmful training samples.
>
> **Real-world experiments with exact Hessian-based influence functions**.
> While we can compute the exact Hessian for the half-moon MLP (a toy model) via second-order automatic differentiation, this already reveals an indefinite and numerically fragile curvature matrix. For real-world networks with millions to billions of parameters, explicitly forming or inverting the exact Hessian is computationally infeasible (both memory and time scale). Therefore, an “exact-Hessian influence” baseline is not a meaningful or practical reference in realistic settings; instead, we focus on scalable influence estimators used in prior work and compare against those established approximations in our real-world experiments.

---

### Review · Reviewer_Jv6o · 2026-02-03

**Summary Of Contributions:**

The authors investigate Hessian-free approximations for Influence Functions (IFs), specifically evaluating the substitution of the inverse Hessian matrix with the identity matrix. They argue that this simplification significantly reduces computational overhead without compromising estimation performance. Furthermore, they propose that ensembling multiple models can further enhance the quality of IF estimations. Beyond traditional data attribution, the authors extend this framework to quantify model behavior in the contexts of fairness and adversarial robustness.

**Audience:**

Yes

**Audience Explanation:**

In the context of data-centric machine learning, identifying high-quality data is a critical task. While Influence Functions (IF) are a powerful data attribution tool, they are often hindered by prohibitive computational costs. This paper addresses these scalability challenges, making it a topic of interest to the broader research community.

**Broader Impact Concerns:**

There are no significant broader impact concerns.

**Claims And Evidence:**

No

**Claims Explanation:**

While the authors claim to “explain the underlying reason” why the Inner Product (IP) outperforms inverse Hessian-based Influence Functions, the provided explanation is somewhat superficial. The paper would benefit from a more rigorous investigation and a deeper analysis to fully substantiate this claim. Specifically, the authors do not provide a rigorous theoretical analysis comparing the two key approximation errors: the gap between the IP and the ideal IF, versus the gap between the estimated inverse Hessian IF and the ideal IF. Without analyzing which error bound is smaller, the claim that IP is superior remains unsubstantiated.

**Requested Changes:**

My primary concern involves the claim that the inverse Hessian can be replaced by the identity matrix. As this substitution represents the central contribution of the paper, I believe a more rigorous and deeper analysis is required to justify its validity.

---

> ### Author Response · Authors · 2026-02-19
> **Response to Reviewer Jv6o**
>
> We thank the reviewer for the thoughtful comments and constructive suggestions. We have revised the manuscript accordingly, and all newly added text is highlighted in blue.
>
> **Hessian matrix**. Thank you for raising this important concern. We agree that the inverse Hessian plays a central role in classical influence-function formulations and thus warrants careful discussion.
>
> In practice, however, it is widely recognized that for modern deep, over-parameterized, and non-convex models, the Hessian is often singular or severely ill-conditioned. As a result, its inverse is either undefined or numerically unstable in practice, even for relatively small neural networks such as the 3-layer MLP used in our toy example. This challenge has been acknowledged by many prior influence-function-based works, which consequently rely on strong approximations, damping, or alternative operators in place of the true inverse Hessian.
>
> Moreover, influence functions are derived from a first-order Taylor approximation, which can already introduce substantial error when applied to highly non-linear deep models. In this regime, even an exact or well-approximated inverse Hessian does not necessarily yield reliable data valuation estimates. Therefore, our use of the identity matrix should not be viewed as an approximation of a well-defined and accurate inverse Hessian, but rather as a deliberate modeling choice that avoids dependence on an ill-posed quantity while providing a stable and computationally efficient estimator.
>
> We acknowledge that some prior works (e.g., DataInf) analyze the approximation error introduced when replacing the inverse Hessian with more tractable operators. While a similar comparison could, in principle, be conducted in our setting, we believe such an analysis would be of limited value. The reason is that the inverse Hessian itself is not a reliable reference in deep, non-convex models. It is well recognized that classical influence functions rely on assumptions that rarely hold in large-scale deep networks. In practice, computing or approximating the inverse Hessian is computationally expensive and can lead to unstable, unbounded, or even misleading estimates. Consequently, measuring the discrepancy between an approximation and the original influence function, such as DataInf, may not provide meaningful insight.
>
> Highlighting these limitations is one of our core motivations. Through this work, we aim to draw attention within the data valuation community to the practical challenges of influence-based methods and to encourage the development of more stable and computationally efficient alternatives. Thus, our paper focuses on empirical robustness, consistency, and scalability, which we view as critical for practical data valuation.
>
> In the revised manuscript, we have also added a dedicated paragraph at the end of Section 3 (titled “When can IP fail?”) that explicitly discusses these limitations and the training conditions under which the gradient-alignment view of IP is expected to be informative, as well as when it can break down in practice. All newly added text is highlighted in blue in the revised version.

---

### Review · Reviewer_PHV7 · 2026-02-19

**Summary Of Contributions:**

Influence functions, an area of research which dates back at least forty-five years in statistics, aim to investigate how much a single data point affects a model's learned parameters. As the authors note, in deep learning, no closed-form solution is guaranteed to exist, and computing the inverse of the Hessian matrix, the most direct approach, is costly. In this work, the authors revisit a Hessian-free method popularized by TracIn, which substitutes the inverse of the Hessian matrix with an identity matrix. They then improve it via ensembling, and conduct experiments on their solution in a variety of settings.

**Audience:**

Yes

**Audience Explanation:**

The machine learning research community has historically shown considerable interest in the study of influence functions, as the authors note. Furthermore, I see no evidence that demand for this area of research is slackening. Therefore, I think some individuals in TMLR's audience will likely be interested.

**Claims And Evidence:**

Yes

**Claims Explanation:**

* The paper provides a lot of experimental evidence in general, which is good, and the idea of getting some beneficial variance reduction via ensembling and dropout is sound and, naturally enough, works well in practice.
* I do not find that, in Section 3, the authors actually explain why "such a naive approximation works well in practice". As the authors note, "choosing the regularization parameter λ presents a tradeoff: a small λ may not guarantee the existence of the inverse, while a large λ makes the inverse approximate the identity matrix." In this case, the theory suggests that Hessian approximations should be better than or the same as IP, unless the effects of approximation error are expected to be worse than the absence of the additional information. To convince otherwise, the authors would have to show that such a large approximation error is likely to occur in practice.
* A similar confusion surrounds the authors' discussion of TracIn. TracIn is very similar to IP, except that the authors consider only the final checkpoint. TracIn does something which, if influence functions behave as expected, should improve the estimation -- namely, by summing contributions at all available checkpoints. Yet the authors write, "While it seems appealing to consider the dynamic sample influence change according to the optimized model parameters, simply summing these simple influence scores along different checkpoints fails to account for the evolving nature of influence, potentially neutralizing conflicting values and reducing overall effectiveness, which can be verified in our experimental results". For a task like noisy label detection, where it mostly matters which samples are detrimental to the final model, perhaps considering only the final checkpoint works best. And indeed, Self-TracIn (which is also single-checkpoint, per Table 1), actually beats plain IP. But it isn't clear why that phenomenon should be expected to generalize to all tasks.

**Requested Changes:**

I think it would be advisable for the authors to consider my comments as to the `accurate, convincing and clear evidence` portion of this review.

P3 typo: calcualtion

---

> ### Author Response · Authors · 2026-02-20
> **Response to Reviewer PHV7**
>
> We thank the reviewer for the thoughtful comments and constructive suggestions. We have revised the manuscript accordingly, and all newly added text is highlighted in blue.
>
> **$\lambda$.** We fully agree with Reviewer PHV7 that a large approximation error is likely to occur in practice. It is difficult to verify this point for a large-scale model. Instead, we found that even in this toy non-linear setting (Figure 1D–F) the exact Hessian is already indefinite and numerically fragile, making “exact-Hessian inversion” an ill-posed and unstable baseline. The minimum eigenvalue is $\lambda_{\min} = -75.85$, confirming the Hessian is indefinite. This observation also suggests that, in such non-convex regimes, spending effort to approximate (and then invert) the true Hessian is arguably not meaningful: the matrix is indefinite and highly sensitive to noise, so small approximation errors can lead to qualitatively different influence estimates.
>
> **TracIn.** While TracIn serves as a competitive baseline in our paper (though not our primary focus), we are happy to elaborate on our understanding of its mechanism. TracIn estimates data influence by accumulating gradient-based contributions across multiple training checkpoints, thereby attempting to capture the dynamic nature of data valuation throughout the training process. We agree that a sample’s influence can vary over time.
>
> However, we argue that a simple summation across checkpoints may not adequately reflect this evolving behavior. For example, a training sample may be highly beneficial at an early checkpoint. After the model updates using that sample’s gradient, its marginal contribution naturally diminishes, since the model has already incorporated its information. Consequently, treating influence as an additive quantity across checkpoints may overestimate its sustained importance. This phenomenon has been discussed in recent work (see Figure 2C in Ref. [1]), which illustrates how data influence can vary substantially during training.
>
> [1] Layer-Aware Influence for Online Data Valuation Estimation.
>
> **Self-TracIn.** Self-TracIn differs from standard TracIn not by using a single checkpoint, but by replacing the validation set with the training set in the influence computation. This modification addresses scenarios where a validation set is unavailable. In general, the validation set is intended to approximate the unseen test distribution; therefore, TracIn computed with a validation set is typically expected to provide more reliable estimates than Self-TracIn, which reuses the training data.
>
> As shown in Table 1, our IP and IP Ensemble consistently outperform both TracIn and Self-TracIn in most individual cases as well as on average, suggesting stronger robustness and practical effectiveness.
>
> **Typo.** Thanks for pointing this out. Fixed.

---

### Decision · Action_Editor_1BMW · 2026-03-11

**Recommendation:** Accept as is

**Audience:**

Yes

**Audience Explanation:**

Influence functions is one standard tool to understand the influence of training samples in resulting predictors. While their use in modern very large models is limited, this paper proposes an efficient methodology for precisely this point.

**Claims And Evidence:**

Yes

**Claims Explanation:**

This paper proposes a simple variation to influence functions relying on a simple identity matrix in replacement with the inverse of the Hessian of a trained predictor in order to conclude on the importance of training samples, suited for modern large and non-convex models. While computationally efficient, the authors demonstrate that this also provides some support on the degree of influence of training samples. The authors further extend this method via ensembles, showing results on fairness and robustness experiments.

All reviewers agree that, after the discussion and revision phases, the paper adequately supports its claims. The exchange between authors and reviewers was productive, and the paper now better explains the lack of importance of the approximation quality of the core component (replacing the Hessian with an identity) in lieu of a better general strategy to gauge sample importance in complex, non-convex settings. Added clarifications and results have also improved the manuscript.

All reviewers recommend acceptance, and I agree.